# Influence Learning in Complex Systems

## Abstract

High sample complexity hampers successful applications of reinforcement learning methods especially in real-world scenarios whose complex dynamics are typically computationally demanding to simulate. One idea is to decompose a large factored problem into small *local* subproblems including only few state variables and model the influence that the external portion of the system exerts on each of them. This principled approach allows to convert the *global* simulator of the entire environment into local lightweight simulators, thus enabling faster simulations, planning and solutions. The ability to represent accurately the influence experienced by each local component is crucial for the effectiveness of this method. In this work, we examine different aspects of the problem of learning approximations of the influence in realistic domains. We empirically investigate several learning methods to conclude that even for large and complex systems, in practice, the influence problem often turns into a relatively manageable learning task. Finally, we discuss how to leverage effectively the influence models for long horizon tasks for planning or reinforcement learning problems. Our results show that in many cases short horizon trajectories collected from the global simulator can be used to obtain accurate approximations of the influence for much longer horizons.

## 1 Introduction

Controlling large distributed systems is a key task in a variety of artificial intelligence fields including computing and information technology (Coulouris et al., 2001), energy systems (Järventausta et al., 2010) and transportation (Dimitrakopoulos & Demestichas, 2010). Reinforcement learning (RL) became a standard framework to study how agents learn and plan in uncertain environments. RL methods showed great promise to tackle large sequential decision making problems (Kaelbling et al., 1998; Sutton & Barto, 2018). However, despite their recent outstanding successes, RL techniques suffer from high sample complexity (Kakade, 2003; Botvinick et al., 2019). In essence, an agent requires a large amount of simulated experience to attain successful performance. Such sample-inefficiency becomes a real hurdle for environments with complex and structured dynamics as in real-world scenarios. In these situations, running expensive simulators to collect a sufficient number of observed samples might be too computationally demanding.

One solution consists of replacing the global model of the entire environment with simpler surrogate representations which enable less expensive simulations. Based on this intuition, influence-based abstraction (IBA) (Oliehoek et al., 2012) provides a principled framework to decompose large multiagent sequential decision making problems into small local subproblems. The IBA approach has shown to lead to significant speed-ups and better performance for RL (Suau et al., 2022b;b) and online planning (He et al., 2020; 2022) problems.

The idea is to leverage the intrinsic factored structure of a broad range of domains to build local models for single agents, including only few relevant factors and abstracting away a large portion of the system. To overcome the implications of the information loss, each of these models is endowed with a representation of the *influence*, which captures the effects of other agent's policies and the *non-local* factors on the local dynamics. This approach allows us to transform a global simulator for the entire environment into many decoupled local simulators. Each one only encompasses few state variables together with the influence that they mutually exert on each other.

In principle, the IBA abstraction process ensures no loss in value: the influence provides a sufficient statistic of the policies of the other agents for one agent's sequential decision making problem to compute optimal solutions (Oliehoek et al., 2021). However, this quality guarantee comes at the cost of introducing a dependency on the history of the local space. In fact, representing the external influence on the subset of local variables corresponds to model the non-Markovian dependencies of local factors. To compensate for the lack of Markovianity, the local model needs to be augmented with histories of local states and actions.

To give a concrete intuition of these concepts, we consider a traffic light control problem for which realistic simulators that can scale to the size of entire cities are usually available. Figure 1 shows a traffic network with a protagonist traffic light agent at intersection 1. Its sensors have the limited capacity to measure the traffic density only in the proximity of the intersection. The red square in Figure 1 delimits a local model for agent 1 which includes all the observed road stretches. We claim that to develop optimal schemes for a single intersection, it is not necessary to collect and store data and reason over the entire large system. For instance, the decisions taken at intersection 3 affect only indirectly the observations of agent 1 through their impact on the car inflows from the north end of the local model (blue arrow). As such, their effects on the sequential decision making problem of agent 1 is captured by the influence of the inflows from the north end. Thus, it suffices to model the non-Markovian dependency of the vehicle streams from the north end on the local traffic and actions. In fact, the abstraction process yields Markovianity breakdown: the car's eastbound outflow, denoted by the red arrow in Figure 1, first impacts the number of cars measured at intersection 2; in turn, this affects the decisions of agent 2; whose results on the inflows in the local model from the north end may be potentially experienced only after several time steps. This means that to predict the north end inflows, the agent needs to reason over the entire history of eastbound outflows. A local model for the protagonist agent should therefore encompass this history.

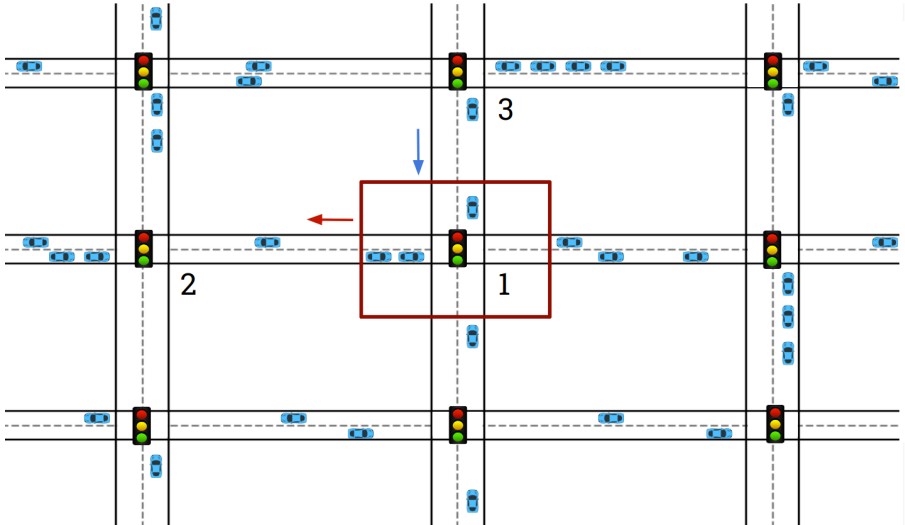

Figure 1: Traffic example. The red square delimits the road segments included in the local model for the traffic light agent 1. The blue arrow represents the influence of the non-local part of the network on the local model and the red arrow the non-Markovian effect of the local traffic on the external part of the system.

The local abstraction enables a significant reduction of the problem at the cost of introducing a (temporal) sequential nature of the local state. This makes analytical computations of the influence or inference unfeasible even for small problems: the space of local histories grows exponentially in time, thus computing exact influence requires solving an exponential number of possibly intractable inference problems. This motivates the idea of learning approximate representations of the influence. We intend to leverage advances in machine learning which have shown excellent outcome in sequence modeling problems (Goodfellow et al., 2016; Gamboa, 2017; Ismail Fawaz et al., 2019) to train approximate models which can generalize the influence over local histories. In this way, we can build a local model characterized by a local state, which in the

traffic example corresponds to all the factors measuring vehicle densities at local level, and a representation of the influence which approximates the car's inflow probability given the past local history. Intuitively, it is crucial to derive sufficiently accurate influence approximations to guarantee a small value loss, thus ensuring near-optimal solutions. A detailed discussion and formal proofs related to this point are provided in Congeduti et al. (2021).

While the use of influence approximations has already proven to yield successes (Suau et al., 2022a; He et al., 2022), it remains unclear to what extend in real-world problems sufficiently accurate representations of influence can be learned and which methods are most suitable for this task. Thus, the first point we intend to address is whether we can train predictors using global simulations to approximate the influence experienced by a small local model in various scenarios and how well different models perform as influence approximators.

As a second step of our investigation, we intend to show that the lightweight local simulators can effectively be employed for solving long horizon tasks. The idea is to demonstrate that the approximate local models learned using short time sequences of experience from the global simulator can be used to sample long trajectories for long horizon planning or RL tasks. In other words, the representations learned using short time experience generalize well the influence over much longer time steps.

The rest of the paper is structured as follows: first the background on influence-based abstraction and the necessary notation are introduced in Section 2; in Section 3, we formulate the formal influence problem and describe the realistic domains used to test our methods; the results on the different models employed as influence predictors are presented in Section 4; we discuss and investigate how to take advantage of influence approximation models for long horizon tasks in Section 5 and finally derive the conclusions in Section 6.

## 2 Background

We consider sequential decision making problems for an agent formulated as Partially Observable Markov Decision Processes (POMDPs) (Kaelbling et al., 1998). Formally, a POMDP is a tuple $\mathcal{M} = (\mathcal{S}, \mathcal{A}, \mathcal{T}, \mathcal{R}, \Omega, \mathcal{O}, b^0, h)$, where $\mathcal{S}$ is the finite space of the states of the environment, $\mathcal{A}$ the space of available actions, $\Omega$ the observation space and $h$ the problem horizon. At the beginning of the process, a state $s^0 \in \mathcal{S}$ is drawn from the initial distribution $s^0 \sim b^0$ [1]. For any discrete time step $t \geq 0$, the agent chooses an action $a^t \in \mathcal{A}$, the state changes according to the distribution $s^{t+1} \sim \mathcal{T}(\cdot \,|\, s^t, a^t)$, the agent receives a reward modeled as $r^t = \mathcal{R}(s^t, a^t)$ and an observation $o^{t+1} \sim \mathcal{O}(\cdot \,|\, s^{t+1}, a^t)$. A policy $\pi$ is a map from action-observation histories $h^t = (a^0, o^1, \dots, a^{t-1}, o^t)$ to probability distributions over action $\pi(h^t) \in \Delta(\mathcal{A})$, which encodes the agent's behavior. The goal of the decision-maker is to optimize the expected cumulative reward for following a policy $\pi$ after a history $h^t$ is observed, $\mathcal{V}^\pi(h^t) = \mathbb{E}\left[\sum_{k=t}^h r^k \,|\, \pi, h^t\right]$. The policy attaining the maximum is denoted by $\pi^*$ and the corresponding optimal value as $\mathcal{V}^*$.

We take the perspective of a single (local) agent $i$ in problems with multiple interacting agents. Precisely, we fix the policies for the external agents $\pi_{-i}$. Then, the best response problem for agent $i$ can be formulated as a POMDP where the variables representing the actions of the other agents $a_{-i}$ become part of the state of the environment. For this reason, we include the actions of the other agents $a_{-i}$ in the state variables of the sequential decision making problem of agent $i$.

We target specific domains whose state space can be decomposed in state variables or *factors*, called factored POMDPs (Hansen & Feng, 2000). This structural property allows to exploit conditional independencies between factors. In particular, it allows to abstract away those factors that have no direct effects on the agent's observations and reward to form an abstract *local model* (Oliehoek et al., 2021). The factors included in the local model, the *local factors*, denoted by $s_{\text{local}}$ retain enough information for the agent to take optimal decisions. The *influence sources* $s_{src}$, are defined as the external state variables (factors and external actions) that directly affect the local factors. All the remaining variables form an external portion of the state discarded in the local abstraction. Accordingly the state can be decomposed as $s = (s_{\text{ext}}, s_{\text{src}}, s_{\text{local}})$.

---

[1] To ease the notation, we use small letters to denote both random variables and their realizations and capital letters to denote sets and functions. For instance, $s^0 \sim b^0$ stands for the random variable representing the state at time 0 is distributed according to $b^0$.

For instance, consider the traffic network introduced in the previous section by Figure 1 from the perspective of the traffic light agent at intersection 1. The state of the space can be thought of as composed by several factors measuring the vehicle densities at different road stretches and the actions of the other traffic light agents. The sensors placed at intersection 1 have the limited capacity to detect the traffic level within a certain range. The aim of the traffic light agent is to minimize the number of vehicles transiting through this local part of the system. Figure 2(a) shows in red two local factors $s_{n\downarrow}$, $s_{w\leftarrow}$ corresponding respectively to the incoming and outgoing traffic flows from the north and west ends of the local model, delimited by the red box. The north end inflow (blue) is an influence source for the local state variables. In Figure 2(b), the two-stage dynamic Bayesian networks (2DBNs) (Boutilier et al., 1999) compactly represent the dependencies between those state variables. The 2DBN on the left side shows that the only external dependency to be modeled to construct well-defined transitions for the local state $s_{\text{local}}^{t+1} \sim \mathcal{T}_{\text{local}}(\cdot \mid s_{\text{local}}^t, s_{\text{src}}^t, a^t)$, is the one on the influence sources (blue arrow). Such dependency encompasses all the non-Markovian effects of local factors on the influence sources: the car's outflow measured by $s_{w\leftarrow}$ has an indirect effect on the future car's inflow measured by $s_{\text{src}}$ throughout its impact on the external state. To compensate for the Markovianity breakdown, the local state is expanded with an history dependent variable which accounts for all the local factors sufficient to infer the influence sources. Formally, this is the set of variables which d-separate the local factors from the external space denoted as $d_{\text{set}}^t \subset \{s_{\text{local}}^0, a^0, \ldots, a^{t-1}, s_{\text{local}}^t\}$. Figure 2(b) shows on the right side, the *influence-augmented local model* (IALM) (Oliehoek et al., 2021), the local abstraction resulting from coupling the observation and reward models with an influence model $I(s_{\text{src}}^t \mid d_{\text{set}}^t) = \mathbb{P}(s_{\text{src}}^t \mid d_{\text{set}}^t)$ (red arrow).

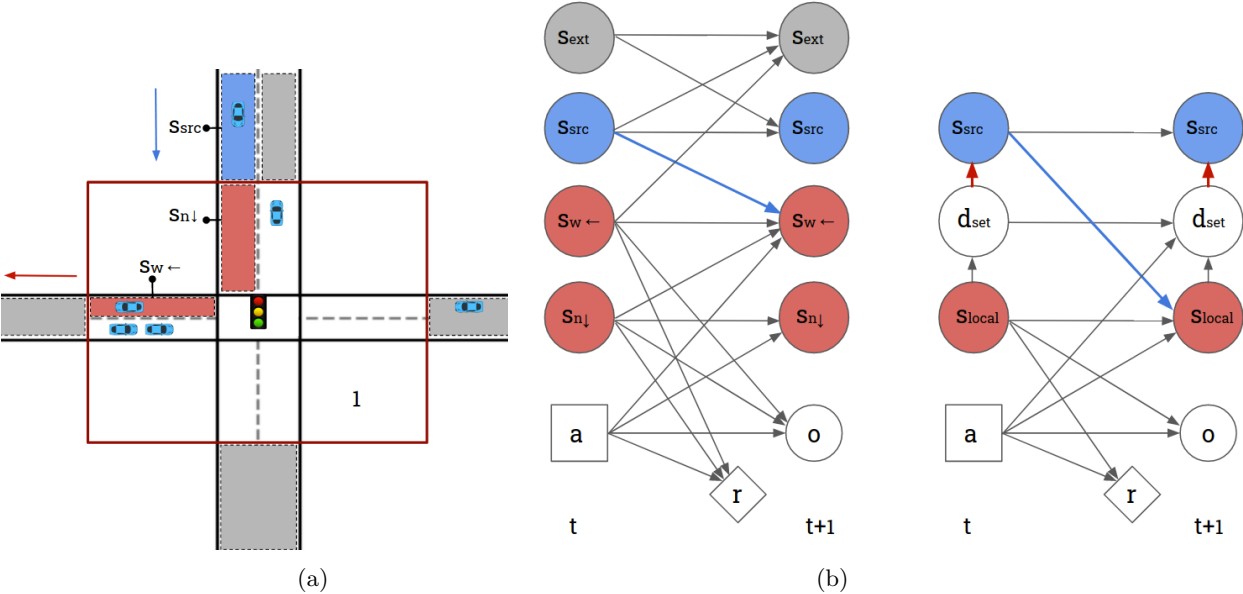

Figure 2: Local model for intersection 1 of the traffic network example. (a) graphical representation of the local state variables and dependencies on external factors; (b) 2DBN representations of the local and external state variables (left) and the abstract IALM (right).

When an exact influence model $I$ is used, the IALM is proven to provide a lossless abstraction: the optimal solutions of the sequential decision making problems represented by the IALM and the original global model coincide Oliehoek et al. (2021). However, computing the influence requires inferring the distribution of the influence sources for each possible instantiation of d-set, whose number grows exponentially with the time. Therefore, inference of the exact influence is typically a computationally intractable problem. One idea to address this issue consists of using a limited number of samples from the global simulator to train learning models that generalize the influence over d-sets. Then, the approximate influence $\hat{I}$ is used to build the transitions for the local simulator $\hat{\mathcal{T}}_{\text{local}}$. The advantage stems from the possibility to retrieve much more samples of experience from the lightweight local simulator to speed-up the solution search. The effectiveness

of this approach heavily relies on the assumption that learning sufficiently good influence approximations is generally an easier task than solving the RL or planning problem. This means that the number of samples required to learn the influence in a first pre-training phase is much smaller than the one needed to learn good policies. This work serves also to provide sufficient empirical evidence that this assumption is in practice well-founded.

## 3 Empirical investigation of influence learning

The first question related to approximating the influence concerns the type of learning task induced. The influence encodes the series of latent Markovian dependencies between local and external variables. Once the external part is abstracted away, such intertwined relations are captured by the non-Markovian dependence of the influence sources on the local state. Therefore, the objective is learning the temporal sequence of distributions of influence sources given the local history. Those are derived by marginalizing over external variable histories whose relations are ruled by an underlying 2DBN structure. As such, the dependencies that the influence learning problem target differ substantially from the benchmark domains typically used to test sequence modeling methods (e.g. speech-recognition, machine translation, audio classification, music generation, image-video caption generation) (Bai et al., 2018; Keneshloo et al., 2019).

Our experimental setup includes a range of realistic scenarios which cover diverse situations in terms of number of influence sources to predict, level of 'uncertainty' of their distributions, strength of the dependency between local variables and influence sources, dependency over past time steps and problem horizon. This leads to different characteristics of the learning tasks. Our intention is to determine if this has an impact on the performance of the learning techniques.

Several learning models can be employed as influence approximators. State-of-the-art methods for sequence modeling are mainly based on neural network with recurrent and temporal convolutions components (Karim et al., 2017; Bai et al., 2018; Ismail Fawaz et al., 2019). The idea of this work is to investigate the variety of influence learning problems along with the most promising learning methods.

### 3.1 Formal framework

Learning influence representations essentially corresponds to predicting the conditional distribution of the influence sources $s_{\text{src}}^t$ given the local factors included in the d-set $d_{\text{set}}^t$ for any $t = 1, \ldots, h$. This is a supervised sequence modeling task: $n$ trajectories of d-sets and influence sources $\mathcal{D} = \{(d_{\text{set}}^{0,i}, s_{\text{src}}^{0,i}), \ldots, (d_{\text{set}}^{h,i}, s_{\text{src}}^{h,i})\}_{i=1,\ldots,n}$ are sampled from the global simulator to train a function approximator $\hat{I}$ for the influence $I(s_{\text{src}}^t \mid d_{\text{set}}^t)$ to minimize the average empirical cross entropy. An approximate influence-augmented local model, $\hat{I}$-IALM, can be defined as $(\mathcal{S}_{\text{local}}, \mathcal{A}, \hat{\mathcal{T}}_{\text{local}}, \mathcal{R}, \Omega, \mathcal{O}, b^0, h)$ over the local space $\mathcal{S}_{\text{local}}$. The $\hat{I}$-IALM inherits the reward and observation functions from the global model. The local transitions $\hat{\mathcal{T}}_{\text{local}}$ are defined by means of the approximate influence $\hat{I}$ by marginalizing over the influence sources.

The idea to leverage the $\hat{I}$-IALM is supported by the intuition that the simulation inaccuracy caused by approximating the local transitions is overcome by the advantages to perform significantly faster local simulations. This is particularly beneficial for RL or planning problems in realistic scenarios. The theoretical insights presented in Congeduti et al. (2021) back up this argument by showing how the value loss for solving the sequential decision making problem in the $\hat{I}$-IALM model is bounded as

$$\mathcal{V}^*(h^t) - \mathcal{V}^{\hat{\pi}^*}(h^t) \leq C \max_t \max_{d_{\text{set}}^t} \sqrt{D_{\text{KL}}(I^t(\cdot \mid d_{\text{set}}^t) || \hat{I}^t(\cdot \mid d_{\text{set}}^t))}, \tag{1}$$

for any history $h^t$. In other words, the difference between the true optimal value $\mathcal{V}^*$ and the value $\mathcal{V}^{\hat{\pi}^*}$ achieved by the $\hat{I}$-IALM optimal policy $\hat{\pi}^*$, is upperbounded by the worst KL-divergence error of the influence predictions over all the possible d-sets, multiplied by a constant $C$ depending only on domain features. This guarantees that any influence approximator, optimizing the mean cross entropy loss $\mathbb{E}_{d_{\text{set}}^t}\left[D_{\text{KL}}(I^t(\cdot \mid d_{\text{set}}^t) || \hat{I}^t(\cdot \mid d_{\text{set}}^t))\right]$ is aligned with the objective of minimizing the value loss $\mathcal{V}^* - \mathcal{V}^{\hat{\pi}^*}$. In fact, even though it does not minimize the maximum itself, it is intuitively clear that in many problems a

low mean implies a low max error. Moreover, this bound suggests that the cross entropy test error provides a priori insight on the value loss. Therefore, it is crucial to show that it is possible to train influence models ensuring good approximations in terms of test error using a small number of training instances.

## 3.2 Experimental domains

In this work different realistic domains have been employed for the empirical investigation. They have been chosen in the attempt to account for the variety of fields of applications and aspects of the influence learning problem arisen from different domain features. Microgrid (MG) represents a realistic application domain for power system management and engineering. We consider a hundred of autonomous units in a power grid interacting by exchanging power with the goal of meeting the energy needs of the different units and minimizing the energy costs. The resulting sequential decision making problem then has a large number of agents. The second domain reproduces the interactions in a Traffic grid (TG) as described in Section 1 and 2. This scenario presents many external factors which exert a direct influence on the local model. Consequently, the influence learning problem imposes high dimensional predictions. System admin (SA) is a simplified version of scenarios in information technology where a team of system administrators needs to cooperate to secure maintenance and operation of computer systems. Given the importance to balance response to short-term or urgent issues and strategies for long-term planning, we set a long horizon for this domain which turns the influence problem into a long-horizon prediction task. Finally, we model a simplified version of the SA as a game called Grab a chair (GC). Given its simplicity, it serves as a controlled scenario for proof-of-concept experiments and preliminary investigations presented in Section 5. This allows us to tune domain parameters such as the stochasticity of the influence sources and the strength of the dependency on the local state, and to test whether these features have an effect on the models performance. Also it serves to assess exactly the performance of the learning models in ad-hoc cases where we know explicitly the analytic expression for the influence. A description of these domains and the resulting influence problems follows below.

**Microgrid (MG).** In this scenario, we model the complex realistic interactions in a power microgrid as a multiagent system (Li et al., 2012; Vlachogiannis & Hatziargyriou, 2004). A hundred of autonomous agents need to manage the energy resources of a microgrid, a small-scale power plant that can operate independently from the rest of the power network. Each agent is responsible for controlling of a single component of the microgrid, a *unit*, which can be a cluster of buildings or a single household unit. They represent residential or industrial consumers that can self-generate and sell power excesses to other units in the grid. In this application, each unit includes a set of renewable sources whose power production per hour is represented by the state variable $P_{RES}$. The energy produced may be employed to meet the hourly power demand $P_d$ or stored in a battery whose state of the charge is represented in percentage by the factor $SOC$. The local model of each agent can be expressed as $s_{\text{local}} = (P_{RES}, P_d, SOC)$. At any time step, corresponding to one hour, the agents may decide also to discharge power from the battery to meet the demand or try to buy/sell energy to neighboring units. When buy and sell orders match, the power from the batteries is exchanged at a small cost/revenue for the buying and selling agents respectively. After these operations, to satisfy the power balance of single units, every agent is forced to buy the residual power demand per hour from an external supplier. Precisely, the agent incurs in a cost $C_{\text{ext}}$ per unit of 'energy not supplied' $\text{ENS} = (P_d - P_{\text{deployed}}) \cdot h$, where $P_{\text{deployed}}$ corresponds to the sum of the renewable power deployed and the power discharged from the battery. The cost $C_{\text{ext}}$ for buying from the external grid is much higher than the fixed operational costs of internal trade $C_{\text{int}}$. A schematic representation of a microgrid unit is depicted in Figure 3(a). The individual reward for each agent is modeled as the sum of the cost/income for the internal trade (if any) and the negative costs of buying the energy not supplied from the external grid (if any) $r = \pm C_{\text{int}} \mathbb{1}_{\text{trade}} - C_{\text{ext}} \text{ENS}$. Thus, the team of agents share the common objective to manage local resources to minimize the electricity costs constrained to satisfying the energy balance, generation limits and storage capacity. The renewable sources at the agent disposal include solar panels and wind turbines. The power output from photovoltaic cells is modeled using hourly solar radiation data provided open source by `https://openweathermap.org/api/solar-radiation`, assuming ideal photovoltaic cells conditions and using the photovoltaic power generation model introduced by Skoplaki & Palyvos (2009). The power output from the wind generator is calculated by transforming the kinetic energy of the wind to electric energy, under

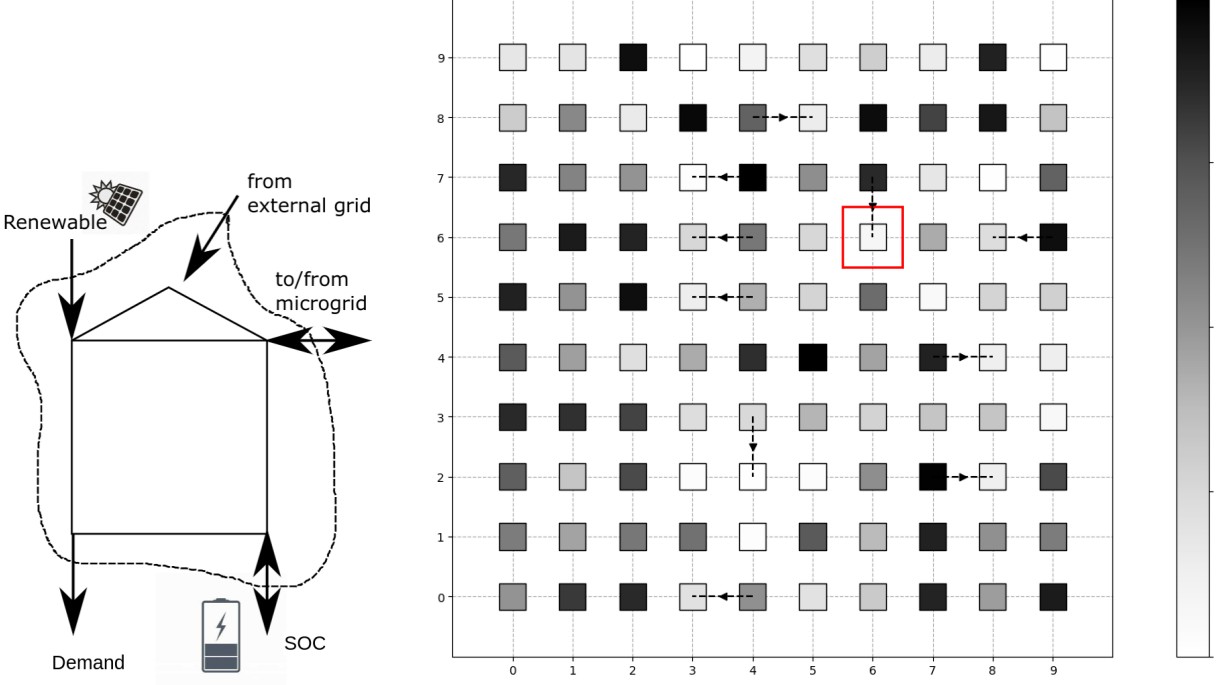

(a) Unit controlled by a single agent in the microgrid. The local variables observed by the agent include the state of the charge, the power produced by renewable sources and the power demand.

(b) Multiagent grid. For each unit in the lattice, the state of charge (in percentage) is represented by the gray scale. Directed edges represent a power exchange from a seller to a buyer agent.

Figure 3: Microgrid.

the assumption that the wind speed follows a Markov chain (Shamshad et al., 2005) and assuming linear relationships with the power produced as explained in details in Kuznetsova et al. (2013). To simulate the dynamics of the demand, we adopt the standard assumption that consumer load follows a normal random distribution Hong & Fan (2016). We take the perspective of a single unit in a lattice network highlighted in red in Figure 3(b). The initial distribution of the battery is uniformly sampled at random and we consider $h = 40$ hours as the horizon of the problem. We assume that all the other agents act by storing or trying to buy power when the storage is scarce and discharging or trying to sell when power is abundant. Note that besides the problem size can be arbitrarily large, the influence experienced by the local agent only directly depends on the neighboring nodes in the network. Precisely, the only relevant information on the external portion of the system that an agent needs is whether the neighboring north, west, south and east agents will decide to sell or buy power. Besides the distributions of influence sources $s_{src} = (a_N, a_W, a_S, a_E)$ are affected (indirectly) by all the agents in the microgrid, the history of local actions $a$ provides a sufficient statistics to predict the influence sources. The resulting problem consists of finding a function approximator for $I(a_N^t, a_W^t, a_S^t, a_E^t \,|\, a^0, \dots, a^{t-1})$.

**Traffic grid (TG)** In this implementation of a traffic network as described in Section 1 and Section 2, we simulate the vehicle traffic in a 9 intersections grid, schematically represented in Figure 4. The sensors of each traffic light capture the vehicles in the $5 \times 5$ local grid at each intersection. The local model is represented as a red square for the selected protagonist agent. The other traffic lights employ hand-coded policies prioritizing lanes with higher car volumes. At time $t = 0$ the grid is initially empty, i.e. the initial state encodes no cars in the network. At any time step, a vehicle will enter the network with a certain probability. The horizon is set to $h = 100$. The state of the environment is represented by binary state

variables detecting the presence/absence of a car in a point of the traffic grid. The goal of the agent is to minimize the total number of vehicles waiting at the local intersection. That is, the reward corresponds to the negative number of cars in the local model. To act optimally, the local agent needs to predict if there will be incoming cars from the north end $s_{n\downarrow}$ and the east end $s_{e\leftarrow}$. Moreover, the local dynamics is affected by traffic congestion at intersection 2 and 4. In fact, traffic jams can prevent vehicles to move out of the local model from the west and south ends. For this reason, the state variables for the outgoing ends and the actions of agents 2 and 4 are included in the set of influence sources. Thus, in addition to the factors encoding cars inflows, the influence sources encompass 4 variables for the west outflow $s_{w\leftarrow}$, 4 variables for the south outflow $s_{s\downarrow}$, the action $a_2$ and $a_4$, that is $s_{\mathrm{src}} = (s_{n\downarrow}, s_{e\leftarrow}, s_{w\leftarrow}, s_{s\downarrow}, a_2, a_4)$. The local information necessary to predict the influence sources includes the entire collection of local variables and actions. The influence that the agent needs to predict is therefore $I(s_{n\downarrow}^t, s_{e\leftarrow}^t, s_{w\leftarrow}^t, s_{s\downarrow}^t, a_2^t, a_4^t \mid s_{\mathrm{local}}^0, a_1^0, \ldots, a_1^{t-1}, s_{\mathrm{local}}^t)$.

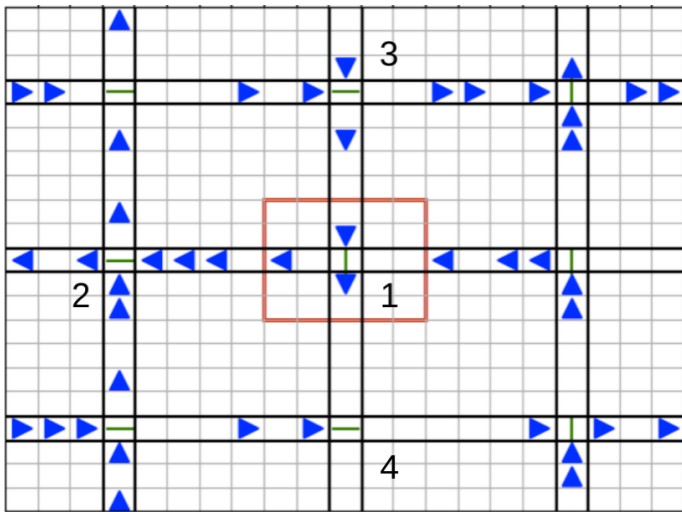

Figure 4: Traffic grid. The local model is delimited by the red square. The blue triangles represent the vehicles and the green bars the traffic lights.

**System admin (SA)** We use a multiagent version of the System administrator domain from Poupart & Boutilier (2004). A team of system administrators are responsible for the upkeep of a network of machines. Each node has a probability of failing at any time step which increases when a neighboring machine in the network is down. Each agent only observes the status of the machines in its proximity. Consequently it may decide to intervene by trying to reboot the system of one of these nodes. With a certain probability the process will succeed resulting in a working node at the next time step. When more than one agent decides to reboot one machine, the process has full rate of success. The goal of the admins team is to secure the highest number of working machines. Precisely, any agent receives a penalty for each faulty machine which lies under its control. We consider a network of $N = 20$ machines organized in a ring configuration as depicted in Figure 5. Each admin agent $i$ is in charge of the maintenance of two neighboring nodes whose state, denoted by $x_i$, $x_{i+1}$, can be fully observed. We take the perspective of a single admin, for instance agent 1 in Figure 5, whose local model includes only the states (faulty or working) of the two neighboring machines $s_{\mathrm{local}} = (x_1, x_2)$ and its action $a_1$. The horizon of the problem is set to $h = 500$ time steps and initially a random state for each machine is sampled. To act optimally, agent 1 needs to know if agents 2 and 20 will decide to reboot one of the two machines over which they share the control. Also, it needs to reason over the neighboring machines status $x_3$, $x_{20}$ as they may contribute to higher the chances to turn down the machines in its local model. Then, according to the influence formalism introduced in Section 2, the sources of influence correspond to $s_{\mathrm{src}} = (x_3, x_{20}, a_2, a_{20})$. The local information at the disposal of the agent 1 to predict the influence sources consists of the entire collection of local variables, i.e. $(x_1, x_2, a_1)$. Thus, the influence problem consists of finding an approximation for the distribution $I(x_3^t, x_{20}^t, a_2^t, a_{20}^t \mid x_1^0, x_2^0, a_1^0, \ldots, a_1^{t-1}, x_1^t, x_2^t)$.

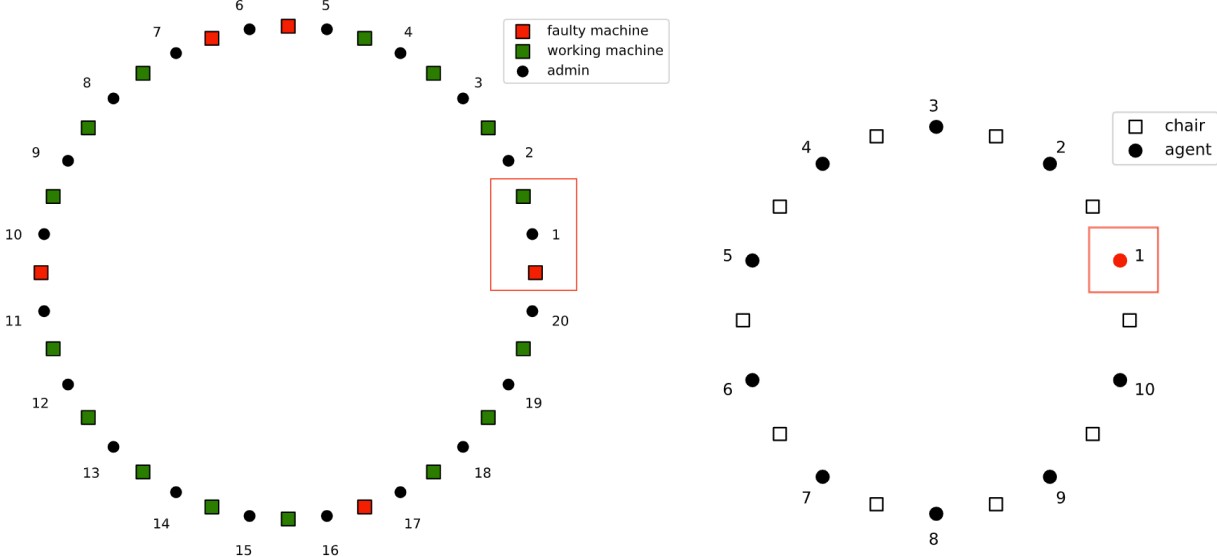

Figure 5: System admin.        Figure 6: Grab a chair.

**Grab a chair (GC)**   In this simplified version of the SA problem, introduced by He et al. (2020), $N$ agents disposed in a ring fashion decide at every time step to grab the chair on their left or right side, as shown in Figure 6. They obtain the chair and thus get the reward only if the neighboring agent has not targeted that chair too. After taking an action, each agent only observes whether it managed to grab the chair, ignoring the action of the neighboring agents. The local agent, numbered by 1 and depicted in red in Figure 6, has no access to other information rather than its own actions and rewards which form the local model. The horizon is set to $h = 200$ and initially every agent chooses deterministically the chair on its right side. After that, all the non-local agents act by copying the previous action of the following agent in counterclockwise order. For the decision making problem of the local agent, the only information required to act optimally consists of the decisions of its neighboring agent 2 and $N$ as they directly influence the possibility to secure a chair. Contrarily, the other agents only affect the local model indirectly. Therefore, the local agent needs to predict the influence sources corresponding to the actions $s_{\mathrm{src}}^t = (a_2^t, a_N^t)$ given the local information of the d-set $d_{\mathrm{set}}^t = (a_1^0, \ldots, a_1^{t-1})$. Therefore the influence to predict corresponds to $I(a_2^t, a_N^t | a_1^0, \ldots, a_1^{t-1})$.

## 4   A comparison of models for learning influence representations

Here we intend to understand how different machine learning models perform when used as influence approximators as explained in Section 3. We aim not only at supporting effective influence-based abstraction methods for reinforcement learning and planning but exploring aspects of the influence learning problems deriving from real-world scale scenarios. In particular, we test the hypothesis that even for large and complex systems, the learning problem may turn into a manageable task that does not require extremely large or overly complex neural networks to be solved. In fact, our results shows that small recurrent or temporal-based networks provide robust, time efficient and accurate approximations of the influence in all the domains used for the empirical validation.

### 4.1   Experimental setup

We consider different learning models ranging from linear models to recurrent and temporal convolutions based models whose performance are evaluated in the System admin, Microgrid and Traffic grid domain. For each of them, we first set the scenario features: number of interacting agents $N$, protagonist agent $i$, policies of the external agents $\pi_{-i}$, problem horizon $h$, initial distribution $b^0$ and the other domain-specific parameters defining the transition and reward functions. The collection of scenarios reflects the diversity of situations that can be encountered in terms of problem size, time-length, influence dimension

and stochasticity. Specifically, the Microgrid represents a large scenario where 100 agents interact in a electric power system domain; in the Traffic grid, a real-world traffic control application turns into a high dimensional learning task; finally the influence learning problem resulting from the System admin implementation is a long-horizon forecasting problem. See Section 3 and Table 5 in Appendix B for the details.

Subsequently, we set a random exploratory policy for the local agent $\pi_i^{\text{Exp}}$ which only affects the data distribution of the training set. Then $n = 500$ trajectories of influence sources and d-sets $\mathcal{D} = \{((d_{\text{set}}^0, s_{\text{src}}^0), \ldots, (d_{\text{set}}^h, s_{\text{src}}^h))_i\}_{i=1,\ldots,n}$ are collected from the global simulator to form training and test sets with a 90% and 10% ratio respectively.

We train different models as approximate influence point $\hat{I} = \{\hat{I}^t(s_{\text{src}}^t|d_{\text{set}}^t)\}_{t=1,\ldots,h}$ using cross entropy as training loss. We consider recurrent models such as Long Short Term Memory (LSTM) (Hochreiter & Schmidhuber, 1997) and Gated Recurrent Unit (GRU)(Cho et al., 2014), temporal convolutions based models such as Temporal Convolutional Network (TCN) (Lea et al., 2017) and Fully Convolutional Network (FullyConv) (Long et al., 2015). Besides state-of-the-art models for sequence predictions, we are interested in evaluating simpler linear models such as a one-layer fully connected linear network, that is, Logistic Regression (LogReg). In order to preserve temporal causality, we employ $h$ independent logistic regression models each one representing the influence for a time step $t \in \{1, \ldots, h\}$. We do not include more complex and larger models such as Transformer (Vaswani et al., 2017; Wen et al., 2022) in our study since we assume and demonstrate that small and simple models can provide good influence representations.

We adopt standard optimization criteria for the networks as ADAM optimization algorithm, linear decay of the learning rate and perform grid search over the space of possible initial and final learning rates. For each scenario a fixed number of epochs and the batch size are chosen. See Table 6 in Appendix B for the hyperparameters configuration of the learning models in each scenario.

To assess the quality of the approximations, we consider the test error $\text{CE}(I, \hat{I})$ computed as the mean over time and sum over influence sources of the cross entropy estimators. Precisely, for each component $s_{\text{src},j}^t$ of the influence sources $s_{\text{src}}^t = (s_{\text{src},1}^t, \ldots, s_{\text{src},J}^t)$ at time $t$, an estimator of the cross entropy loss $\hat{\text{CE}}(P(s_{\text{src},j}^t \mid d_{\text{set}}^t), \hat{P}(s_{\text{src},j}^t \mid d_{\text{set}}^t))$ can be computed over the test sample as

$$\hat{\text{CE}}(P(s_{\text{src},j}^t \mid d_{\text{set}}^t), \hat{P}(s_{\text{src},j}^t \mid d_{\text{set}}^t)) = -\frac{1}{n}\sum_{i=1}^{n} \ln(\hat{P}(s_{\text{src},j,i}^t \mid d_{\text{set},i}^t)). \tag{2}$$

The resulting approximate influence is given by the probability distributions $\hat{I}^t(s_{\text{src}}^t \mid d_{\text{set}}^t) = (\hat{P}(s_{\text{src},1}^t \mid d_{\text{set}}^t), \ldots, \hat{P}(s_{\text{src},J}^t \mid d_{\text{set}}^t))$. Then the error $\hat{\text{CE}}(I^t, \hat{I}^t)$ for each time step $t$ is defined as the sum over the errors over each component of the influence sources

$$\hat{\text{CE}}(I^t, \hat{I}^t) = \sum_{j=1}^{J} \hat{\text{CE}}(P(s_{\text{src},j}^t \mid d_{\text{set}}^t), \hat{P}(s_{\text{src},j}^t \mid d_{\text{set}}^t)), \tag{3}$$

and the test error is obtained by averaging the errors over time

$$\text{CE}(I, \hat{I}) = \frac{1}{h}\sum_{t=1}^{h} \hat{\text{CE}}(I^t, \hat{I}^t). \tag{4}$$

We also measure the wall-clock training times (WCTTs) to determine which model provides the best trade off between error an time.

## 4.2 Results

The models have been trained over 15 epochs for the Microgrid and 20 epochs for Traffic grid and the System admin. During each epoch, the training set is fed to the network in batches of 100 trajectories. Standard optimization criteria are adopted as summarized in Table 6 in Appendix B. For each domain, we compare the results of networks with different sizes. See Table 7, 8 and 9 in Appendix B for the details of the architectures used for Microgrid, Traffic grid and System admin respectively.

| models | number of parameters | | | number of parameters | | |
|---|---|---|---|---|---|---|
| | $\leq 100$ | 1000 | 15000 | $\leq 100$ | 1000 | 15000 |
| LSTM | $3.81 \pm 0.03$ | $3.78 \pm 0.02$ | $3.82 \pm 0.03$ | 0.98 | 1.18 | 2.26 |
| GRU | $3.77 \pm 0.03$ | $3.73 \pm 0.03$ | $3.78 \pm 0.03$ | 0.95 | 1.13 | 2.15 |
| TCN | $4.11 \pm 0.03$ | $3.98 \pm 0.03$ | $3.85 \pm 0.04$ | 0.47 | 0.59 | 1.31 |
| FullyConv | $3.85 \pm 0.03$ | $3.85 \pm 0.03$ | $3.76 \pm 0.03$ | 0.46 | 0.65 | 1.55 |
| LogReg | | - | $3.85 \pm 0.04$ | - | - | 0.95 |

(a) Cross Entropy test error.   (b) Wall-clock training time (s).

Table 1: Microgrid, means and standard errors over 10 iterations of the experiment.

### 4.2.1 Microgrid

Table 1a lists the test errors computed according to equation 4 for different network sizes averaged over 10 runs of the experiment with the corresponding standard errors. Table 1b reports the respective average

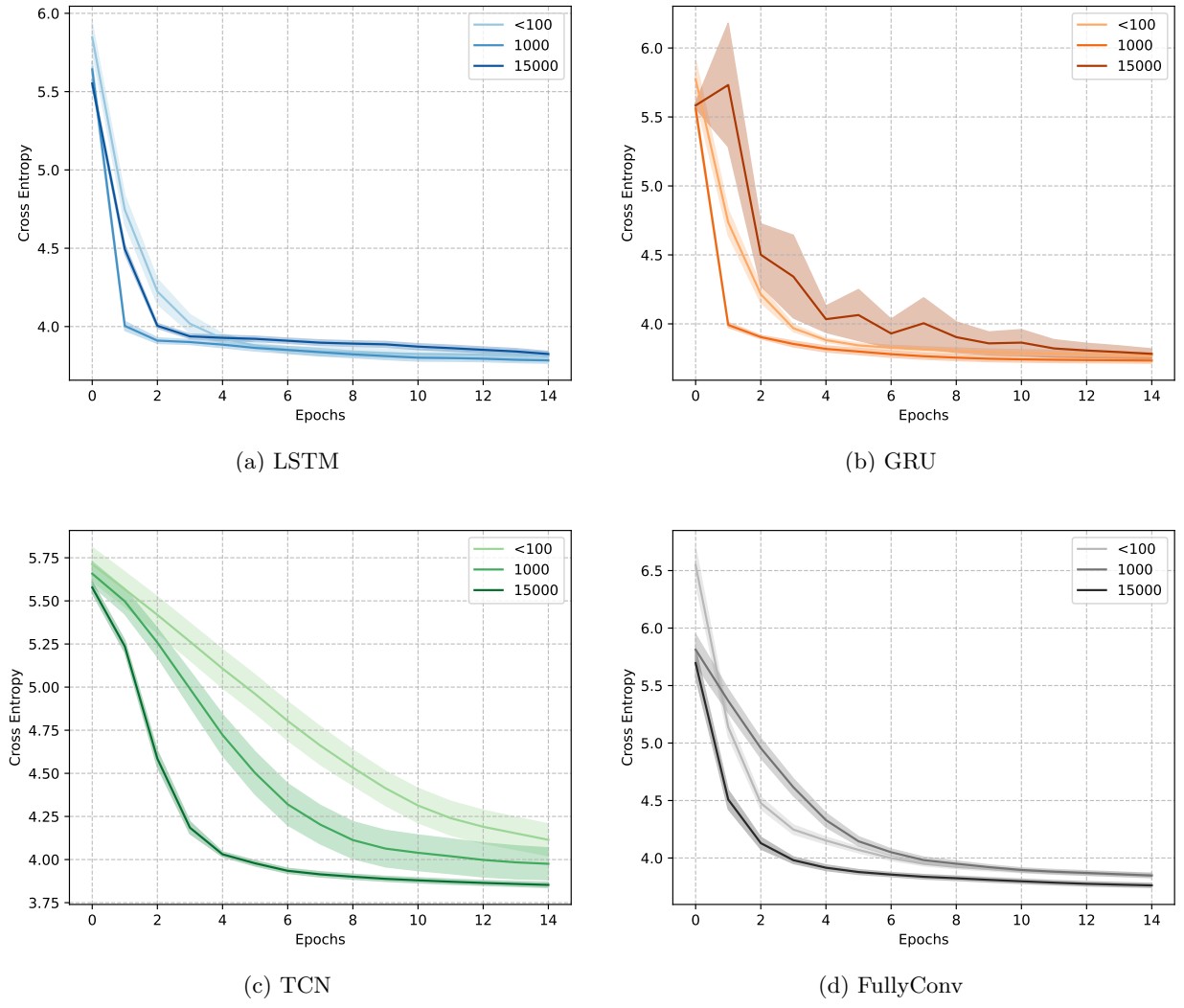

(a) LSTM

(b) GRU

(c) TCN

(d) FullyConv

Figure 7: Microgrid. Cross entropy test errors over epochs.

WCTTs. We omit the standard errors because they are negligible in the training time scale. The tested sizes range from very small networks (less than 100 parameters) scaling up to the number of parameters of a one-layer linear neural network (LogReg). The test errors show no significant advantages from larger size of the networks with the only exception of the temporal convolutional network. In particular, the recurrent based models do not benefit from larger sizes, as shown in Figure 7(a) and 7(b). The FullyConv shows a slight improvement on the performance at the cost of tripling the training time (see Table 1b and Figure 7(d)). A different situation is recorded for the simple temporal convolutional network: increasing the size significantly improve the performance (see Figure 7(c)).

In general, temporal convolutional networks require more layers to have full receptive fields, i.e. to ensure that the network processes the entire history-length of the input sequence to output the predictions. Also, recurrent layers tend to have fewer parameters than temporal layers (Bai et al., 2018). Nevertheless, the higher amount of memory required to store the network parameters is compensated by the lower training times.

For each model, we consider the Pareto front (Miettinen, 1999) for the multi-objective optimization problem with test error and training time as objective functions. As an example, Figure 8(a) shows the Pareto frontiers for some models: the points represent test error and training time for one choice of the network size; the Pareto fronts represented by the dashed lines include all the sizes which are not strictly dominated by any other. Among the Pareto optimal solutions, we select and further analyze specific sizes which seem to provide a good trade-off between the error and training time (red points in Figure 8(a)). The performance measure of those models are highlighted in the gray cells in Table 1. In Figure 8(b), we collect the learning curves of the selected models. Overall, the recurrent models, the fully convolutional network and the logistic regression show similar performances: the small accuracy loss of the FullyConv compared to the recurrent models is compensated by the lower training times. The logistic regression attains the same performance levels of the non-linear models with a much larger number of parameters. One explanation for these results is that the influence learning problem for the Microgrid domain is a relatively simple learning task. In fact, the problem is low dimensional and presumably the sources of influence depend only loosely on the local history of the agent. Remarkably, this is a representative situation for influence problems in many complex scenarios: the local problems are generally sufficiently well-decoupled from the rest of the system, i.e. the local variables affect only weakly the influence source distributions. As such, even simple or non sequence-based learning models have the potential to learn good influence approximations. In this specific scenario, even expanding the microgrid network to account for an arbitrarily large number of units, the influence problem would result in the same small dimension learning task.

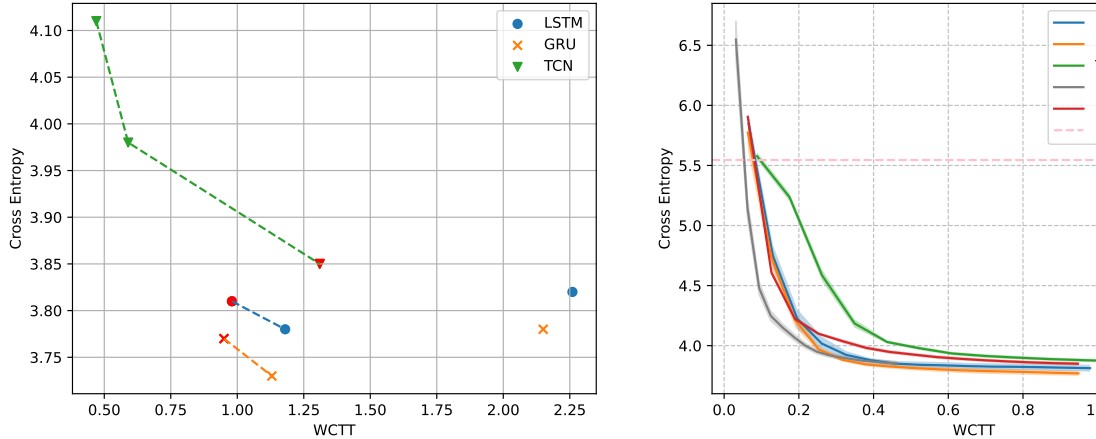

(a) Pareto fronts. The sizes selected are represented in red and correspond to the cells highlighted in Table 1.

(b) Test error over wall-clock training time for the selected sizes.

Figure 8: Microgrid.

| **models** | **number of parameters** | | | | | **number of parameters** | | | | |
|---|---|---|---|---|---|---|---|---|---|---|
| | $\leq 200$ | 1K | 10K | 50K | 1M | $\leq 200$ | 1K | 10K | 50K | 1M |
| LSTM | $6.2 \pm 0.4$ | $3.4 \pm 0.3$ | $3.1 \pm 0.3$ | $3.2 \pm 0.3$ | - | 20.1 | 20.9 | 25.4 | 40.2 | - |
| GRU | $6.6 \pm 0.6$ | $3.3 \pm 0.2$ | $3.1 \pm 0.3$ | $3.2 \pm 0.3$ | - | 21.4 | 22.2 | 27.5 | 40.6 | - |
| TCN | $7.4 \pm 0.2$ | $6.6 \pm 0.0$ | $4.3 \pm 0.4$ | $5.5 \pm 0.5$ | - | 6.0 | 7.4 | 8.2 | 14.3 | - |
| FullyConv | $6.3 \pm 0.2$ | $4.4 \pm 0.3$ | $3.1 \pm 0.3$ | $3.1 \pm 0.3$ | - | 6.5 | 7.9 | 10.3 | 17.0 | - |
| LogReg | - | - | - | - | $5.6 \pm 0.3$ | - | - | - | - | 35.3 |

(a) Cross Entropy test error.          (b) Wall-clock training time (s).

Table 2: Traffic grid, means and standard errors over 10 iterations of the experiment.

### 4.2.2 Traffic grid

The test errors and training times in Table 2 show that models with very small numbers of parameters are unable to attain good performance. In fact, the higher dimensionality of the space of influence sources and

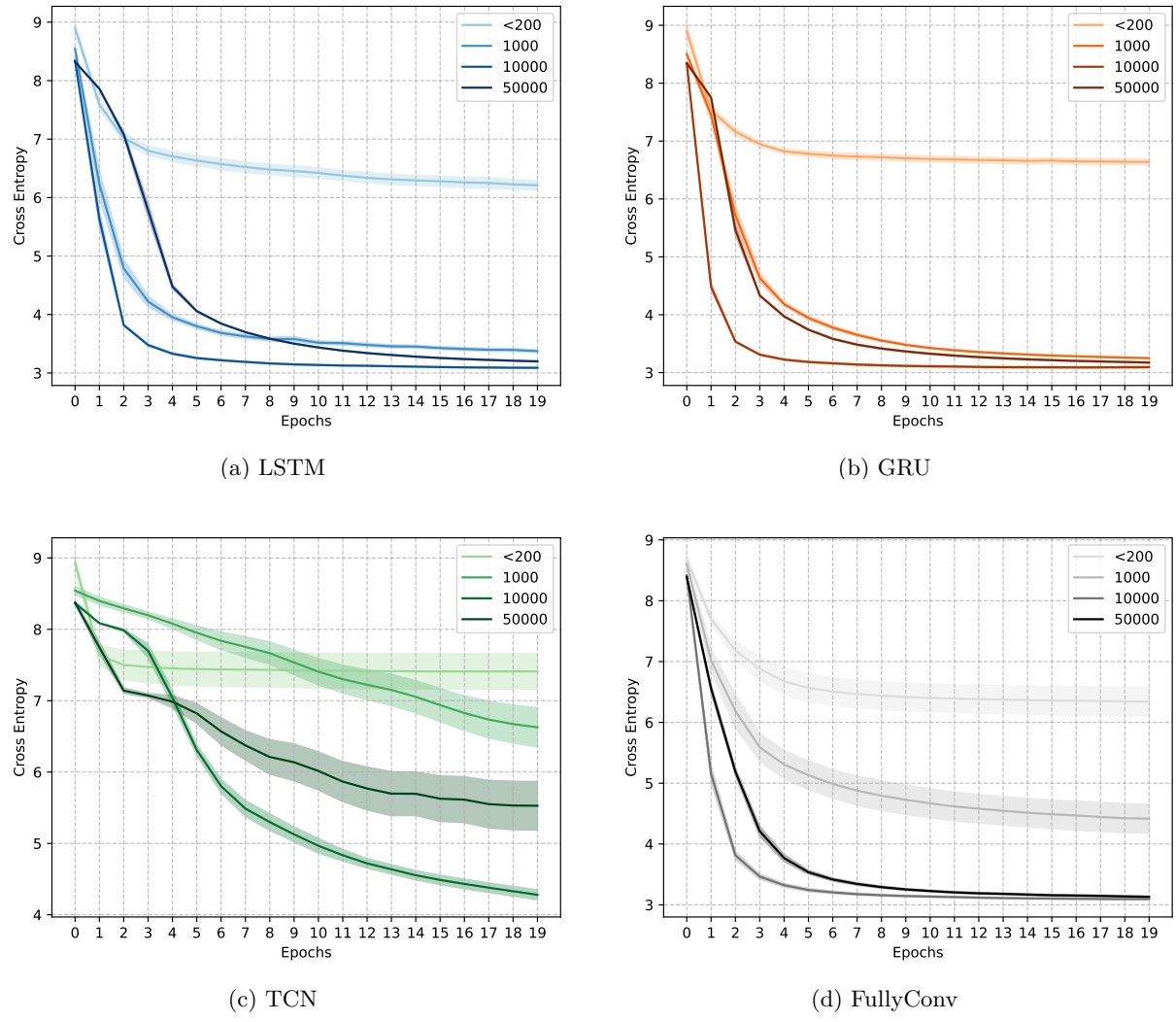

(a) LSTM                    (b) GRU

(c) TCN                    (d) FullyConv

Figure 9: Traffic grid. Cross entropy test errors over epochs.

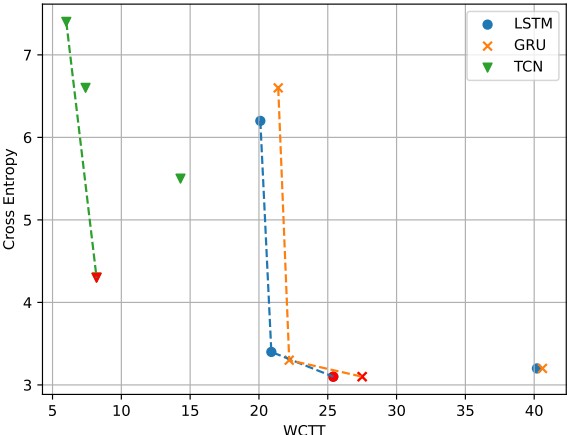

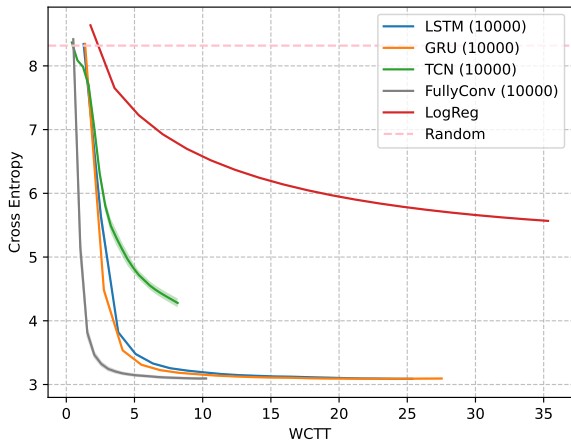

(a) Pareto fronts. The sizes selected are represented in red and correspond to the cells highlighted in Table 2.

(b) Test error over wall-clock training time for the selected sizes.

Figure 10: Traffic grid.

d-sets increases the complexity of the learning problem. As in the previous domain, we select the Pareto optimal sizes for which the models achieve a reasonable trade-off between test error and training time (see Figure 10(a)). Notice that larger sizes do not result into lower test errors and raise the training times. This suggests that even for higher dimensional problems, relatively small architectures are preferable and learning the influence still remains a sufficiently simple task. In particular, Figure 9 shows that recurrent models (LSTM, GRU) and FullyConv network display similar behavior: small networks do not have enough capacity to accomplish the learning task while exceeding the number of 10000 parameters has the mere result to deteriorate the performance and prolong the training times. Also the temporal convolution network does not benefit from more than 10000 parameters. The learning curves in Figure 9(c) show that the TCN model presumably needs more training steps to converge. We compare the test errors over training time for the sizes selected for each model in Figure 10(b). In this case, the linear model fails to reach the accuracy levels attained by recurrent models and FullyConv. Moreover, the TCN shows its limitations in more complex problems: even if the model has not converged yet, the learning curve shows a significantly slower learning speed compared to recurrent and fully convolutional models.

### 4.2.3 System admin

In this long horizon scenario ($h = 500$), all the models achieve good performance in terms of test error and training time for relatively small network sizes as shown in Table 3. As in the previous scenarios, the models based on temporal convolutions seem to benefit more from larger sizes than the recurrent models. For this

| | number of parameters | | | | number of parameters | | | |
|---|---|---|---|---|---|---|---|---|
| **models** | $\leq 100$ | 1K | 10K | 3M | $\leq 100$ | 1K | 10K | 3M |
| LSTM | $1.69 \pm 0.04$ | $1.06 \pm 0.04$ | $1.05 \pm 0.04$ | - | 13.7 | 18.5 | 32.4 | - |
| GRU | $1.68 \pm 0.04$ | $1.06 \pm 0.04$ | $1.05 \pm 0.04$ | - | 13.1 | 18.3 | 33.9 | - |
| TCN | $2.06 \pm 0.05$ | $1.98 \pm 0.03$ | $1.05 \pm 0.04$ | - | 3.6 | 5.1 | 12.7 | - |
| FullyConv | $2.01 \pm 0.04$ | $1.20 \pm 0.05$ | $1.05 \pm 0.04$ | - | 3.9 | 5.2 | 12.3 | - |
| LogReg | - | - | - | $1.97 \pm 0.08$ | - | - | - | 23.2 |

(a) Cross Entropy test error.

(b) Wall-clock training time (s).

Table 3: System admin, means and standard errors over 10 iterations of the experiment.

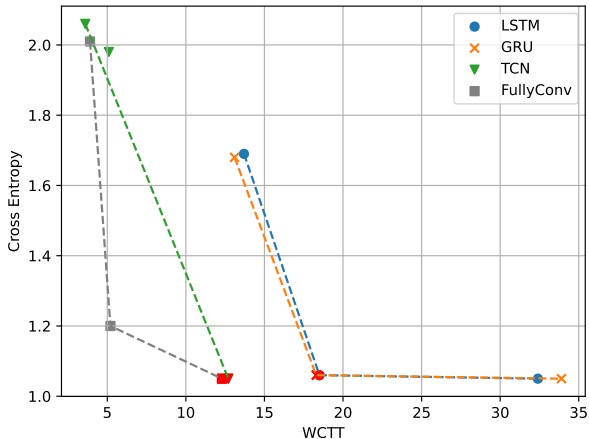

(a) Pareto fronts. The sizes selected are represented in red and correspond to the cells highlighted in Table 3.

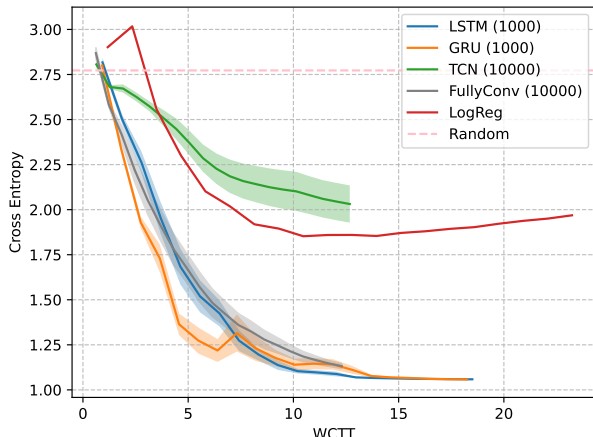

(b) Test error over wall-clock training time for the selected sizes.

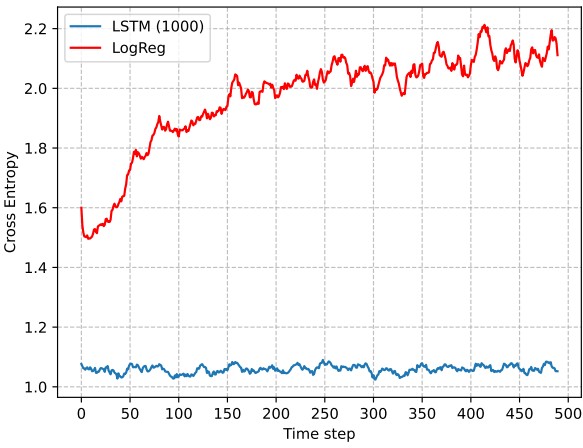

(c) Test error per time steps.

Figure 11: System admin.

reason, we select larger Pareto optimal sizes for the convolutions based models than for recurrent models (see Figure 11(a)). A comparison of learning curves of different network sizes is also shown in Figure 15 in Appendix A. Figure 11(b) shows that LSTM, GRU and FullyConv are the best suited models for this problem with similar learning speed and final accuracy.

The logistic regression shows its limitation in representing long-term temporal dependencies. To better understand this flaw we plot in Figure 11(c) the test error over each time step of the logistic regression compared to the LSTM. The figure shows that the logistic regression error increases over time steps while the LSTM error remains constant. The fact that the predictions become more inaccurate for later times means that the capability of the logistic regression to represent the influence decreases with the problem horizon. One explanation for this is that a linear model is not able to accurately represent long temporal dependencies between the local variables and the influence sources as they become increasingly non-linear with the growth of the problem horizon.

#### 4.2.4 Observations

We can summarize the main conclusions of the empirical investigation of this section as follows:

- Complex realistic scenarios typically induce manageable influence learning task which relatively small recurrent models can effectively handle.

- Learning models not designed to model temporal dependencies do not provide adequate solutions to represent the influence.

- Recurrent models have been demonstrated to perform equally as well as state-of-the-art architectures for modeling temporal sequences, such as fully convolutional neural networks, and proved to be the most suitable models to represent the influence.

## 5 Approximate influence for long horizon tasks

In many real-world applications, our influence predictors would operate for very long time. Some examples include sequential decision making problems that require long horizon samples or might substantially benefit from them as in long horizon planning (Simeonov et al., 2020; Pertsch et al., 2020), episodic reinforcement learning (Dann & Brunskill, 2015) and sparse reward tasks (Riedmiller et al., 2018). In the previous sections we discussed how approximate influence-based abstraction can speed up simulations in large realistic scenarios: learning models can effectively approximate the influence at the negligible cost of running global simulations to collect a manageable training sample. This may no longer be the case when dealing with long planning horizons. In fact, collecting long trajectories from the global simulator might still be computational demanding. One idea is to leverage learning models to generalize the influence representations beyond the training horizon. With this respect, the question we want to answer is to what extent the representations learned using trajectories with horizon $h_{\text{train}}$ can approximate well the influence for a much longer deploying horizon $h_{\text{deploy}} \gg h_{\text{train}}$. The relevance of this question lies in the possibility to leverage these approximations to collect much longer trajectories by using lightweight local simulators.

This intuition stems from the Ergodic Theory for MDPs (Puterman, 2014; Morton & Wecker, 1977; Kearns & Singh, 2002). Under ergodic assumptions, the Markov chain induced by the joint policy $\pi = (\pi_1, \ldots, \pi_N)$ converges to the stationary distribution regardless of the initial distribution, at a rate depending on the mixing time of the system. This suggests the idea of a *stationary influence point* as the limit conditional distribution function of the sequence $\{I^t\}_t$. That is, when the system is sufficiently mixed the influence resulting from $\pi$ will approach a time-independent function of few factors in the $d_{\text{set}}$, representing the conditional distribution of the influence sources at equilibrium. We would like to emphasize that the concept of a steady-state behavior of the influence relies merely on heuristic arguments based on ergodicity for Markov systems, and is further endorsed by the empirical observations gathered in this section. These arguments support the intuition that learning models can represent well the influence for decision making problems with long horizon $h_{\text{deploy}}$. More precisely, assuming that the influence tends to a stationary distribution, global simulator trajectories with horizon longer than the mixing time $h_{\text{train}} > t_{\text{mix}}$, contain sufficient information to learn the stationary influence point. Thus, approximators trained over relatively short trajectories, have the potential to generalize well the influence over a (indefinitely) longer horizon $h_{\text{deploy}}$.

This raises a key question: which is a suitable training horizon $h_{\text{train}}$ for such task? Clearly, we expect that the influence predictions improve for training horizons tending to the deploying horizon $h_{\text{train}} \to h_{\text{deploy}}$. However, the intent is to limit the computational effort from running expensive global simulations for long times. Thus, we seek a target training horizon $h_{\text{train}}^*$ as the minimal number of time steps which guarantees accurate influence predictions for a long deploying horizon. Also, this target horizon needs to ensure negligible performance loss when compared with models trained over longer sequences. In essence, the choice of a training horizon $h_{\text{train}}^*$ is a trade-off between the quality of approximations and the length of the training sequences. Clearly, a good candidate $h_{\text{train}}^*$ is domain specific: it essentially boils down to the sum of the system mixing time and some additional time steps necessary for the learning model to generalize the experience encountered after mixing and capture the steady state. Thus, in principle, $h_{\text{train}}^*$ depends also

on the learning model itself. However, we empirically show that this dependency is weak when comparing models that have demonstrated to be effective for the influence learning task.

## 5.1 Experimental setup

We consider a set of $K$ training horizons $H = \{h_1, \ldots, h_K\}$ with $h_1 \leq \cdots \leq h_K$. The index $k \in \{1, \ldots, K\}$ identifies the choice of a training horizon $h_{\text{train}}$ in the set $H$. We collect $n$ global simulator trajectories of d-sets and influence sources with horizon $h_K$, assuming a random exploratory policy for the local agent $\pi^{\text{Exp}}$. Then, we train $K$ different learning models such that the $k$-th model learns the influence approximation $\hat{I}_k$ using as training set the truncations of the trajectories up to horizon $h_{\text{train}} = h_k$, $\mathcal{D}_k = \{((d_{\text{set}}^0, s_{\text{src}}^0), \ldots, (d_{\text{set}}^{h_k}, s_{\text{src}}^{h_k}))_i\}_{i=1,\ldots,n}$.

To assess the ability of each of the $K$ models to capture the influence for long horizons, we test them over longer trajectories. An independent test sample with horizon $h_{\text{deploy}} \gg h_K$ is used for this purpose, $\mathcal{D}_{\text{deploy}} = \{((d_{\text{set}}^0, s_{\text{src}}^0), \ldots, (d_{\text{set}}^{h_{\text{deploy}}}, s_{\text{src}}^{h_{\text{deploy}}}))_i\}_{i=1,\ldots,m}$. We define the error function $\text{e}^k(h)$, representing the generalization error of the $k$-th influence model over $h$ deploying time steps. That is,

$$\text{e}^k(h) = \frac{1}{h} \sum_{t=1}^{h} \hat{\text{CE}}(I^t, \hat{I}_k^t), \tag{5}$$

where $\hat{\text{CE}}(I^t, \hat{I}_k^t)$ is the empirical estimate of the cross entropy between ground true $I^t$ and predicted $\hat{I}_k^t$ influence computed over the test set $\mathcal{D}_{\text{deploy}}$ according to equation 3. Note that for $h = h_{\text{deploy}}$, $\text{e}^k(h_{\text{deploy}})$ corresponds to the average error of the influence model trained using $h_k$ horizon trajectories over the independent test sample for $h_{\text{deploy}}$ time steps. As such, it measures how well (on average) the $k$-th model generalizes the influence over the entire deploying horizon. While for $h = h_k$, $\text{e}^k(h_k)$ represents the error of model $k$ computed as the average error over the independent test trajectories with same length $h_k$ as the training trajectories. Precisely, the *test error* and *deploying error* of the $k$-th model are defined as

$$\text{e}_{\text{test}}^k := \text{e}^k(h_k) \tag{6}$$
$$\text{e}_{\text{deploy}}^k := \text{e}^k(h_{\text{deploy}}) \tag{7}$$

Our aim is not limited to investigate an advantage in terms of computational time but also to estimate the deploying error by using the short horizon trajectories. Intuitively, the test error should be a good candidate to fulfill this task. However, when the training horizon is short compared to the deploying horizon, the cross entropy errors $\hat{\text{CE}}(I^t, \hat{I}_k^t)$ before mixing $t \leq t_{\text{mix}}$ affect substantially the test error. In general, those error terms for initial time steps deviate significantly from those after stationarity is reached. Therefore, the test error often fails in representing the deploying error whose main contribution is given by error terms after mixing. One solution to mitigate this mismatch, is to neglect the non-informative errors in early time steps: the losses before mixing do not add any information on the long term model performance. On the other hand, after mixing we expect learning models to form an approximately steady representation of the stationary influence point. As such, the errors $\hat{\text{CE}}(I^t, \hat{I}_k^t)$ for $t > t_{\text{mix}}$, may represent well the generalization ability of the model. Thus, the idea is to use a window of $l$ time steps of the global trajectories to define the *test-tail error* as

$$\text{e}_{\text{tail}}^k = \frac{1}{l} \sum_{t=h_k+1}^{h_k+l} \hat{\text{CE}}(I^t, \hat{I}_k^t). \tag{8}$$

This means that for training horizon $h_k$, $l$ time steps of experience in the test set $\{((d_{\text{set}}^{h_k+1}, s_{\text{src}}^{h_k+1}), \ldots, (d_{\text{set}}^{h_k+l}, s_{\text{src}}^{h_k+l}))_i\}_{i=1,\ldots,m} \subset \mathcal{D}_{\text{deploy}}$ are used to evaluate the model. We empirically demonstrate that for $h_k \geq t_{\text{mix}}$, the error $\text{e}_{\text{tail}}$ offers a better estimate of $\text{e}_{\text{deploy}}$ compared to $\text{e}_{\text{test}}$. Thus, the test-tail error can be used to assess a priori the quality of the model predictions for long deploying horizons and search for an optimal training horizon $h_{\text{train}}^*$.

An LSTM and a fully convolutional network are used for the experiments since they have proven to achieve good performance in the influence learning task (see the results in Section 4.2). For each scenario, the hyperparameter setting has been chosen ensuring that both models share the same size. See Table 12 in Appendix B for the details of the network parameters.

We use for the experiments the domain Grab a chair as described in Section 3.2 which serves as controlled environment where a known stationary influence point is reached after few time steps. Precisely, we assume that all external agents $2, 3 \ldots, N$ act by copying the action of the following agent in counterclockwise order (see Figure 6). For instance, agent 3 at time $t$ copies the last action of agent 2, which is, in turn, the action of the local agent 1 at time $t - 2$. In general, for any agent $i$, $a_i^t = a_1^{t-i+1}$. We assume that initially all the agents pick deterministically the chair on their right side and then copy the neighbor's previous action. Then, after $N - 1$ time steps the system converges to a stationary distribution: any agent's action is distributed according to the local agent policy, that is $a_i^t \sim \pi_1^{\text{Exp}}$ for $t \geq N - 1$. As a result, the influence is a time independent function of the last $N$ local actions and deterministically determined by

$$I(a_2^t, a_N^t | a_1^0, \ldots, a_1^{t-1}) = \mathbb{P}(a_2^t, a_N^t | a_1^{t-1}, a_1^{t-N+1}) = \delta_{a_1^{t-1}}(a_2^t)\delta_{a_1^{t-N+1}}(a_N^t) \qquad \text{for } t \geq N - 1. \qquad (9)$$

Such closed formula for the influence allows us to analyze the model learning for the different training horizons. We consider a first scenario (GC4) with $N = 4$ interacting agents and a second with $N = 11$ agents (GC11). We also use the Traffic grid (TG) domain as described in Section 3.2 to validate our assumptions in a more realistic environment where no prior knowledge on stationarity is available. See Table 10 in Appendix B for the details of the scenario settings.

## 5.2 Results

The models have been trained over 25 epochs for GC4, 30 epochs for GC11 and 20 epochs for TG. During each epoch, the training set is fed to the network in batches of 100 trajectories. For each domain, we compare the results over different training horizons. Specifically, for GC4 the learning models have been trained for $h_{\text{train}} \in H = \{2, \ldots, 14\}$ and tested for $h_{\text{deploy}} = 200$. For GC11, $h_{\text{train}} \in H = \{5, \ldots, 30\}$ and $h_{\text{deploy}} = 200$. In TG, $h_{\text{train}} \in H = \{5, \ldots, 100\}$ and $h_{\text{deploy}} = 500$. Table 11 in Appendix B specifies the optimization choices for the learning models.

### 5.2.1 Generalization beyond training horizon

Table 4 lists the errors $\text{e}_{\text{deploy}}$ averaged over 10 iterations with the corresponding standard errors for a given choice of the training horizon $h_{\text{train}}$ in the three scenarios. The performance of the models are compared with the error of a random classifier. The choice of a suitable training horizon is domain-dependent: in the Grab a chair scenario with 4 agents, $h_{\text{train}} = 6$ training steps are sufficient to get deploying error close to 0 while for 11 agents, the models require a longer training horizon $h_{\text{train}} = 22$. Such choices have been driven by idea that the training horizons need to be longer than the mixing time $t_{\text{mix}}$, which corresponds respectively to $t_{\text{mix}} = 3$ and $t_{\text{mix}} = 10$. The results show that a training horizon slightly longer than the mixing time ensures cross entropy errors close to zero for much longer times. In other words, few training time steps of experience after mixing are sufficient for the models to generalize the deterministic stationary influence over 200 deploying time steps. For the Traffic grid the results are less straightforward to interpret. In fact, the cross entropy error depends on the entropy of the target influence distributions, which are unknown. However, for $h_{\text{train}} = 30$ the average errors over 500 deploying time steps in Table 4 are significantly lower than the random classifier error. Also, they are quite close to the errors computed over a much shorter test horizon and reported in Table 2a. This leads to conclude that 30 training steps are sufficient to learn good long term influence approximations. In summary, for every scenario there exist horizons $h_{\text{train}}$ sufficient to train influence models that can approximate well the influence for $h_{\text{deploy}}$ time steps ensuring small deploying error.

| | learning model | | | | |
| scenario | LSTM | FullyConv | Random | $h_{\text{train}}$ | $h_{\text{deploy}}$ |
|---|---|---|---|---|---|
| GC4 | $0.002 \pm 0.002$ | $0.027 \pm 0.026$ | 1.38 | 6 | 200 |
| GC11 | $0.002 \pm 0.001$ | $0.025 \pm 0.01$ | 1.38 | 22 | 200 |
| TG | $3.68 \pm 0.06$ | $3.98 \pm 0.07$ | 8.32 | 30 | 500 |

Table 4: Mean and standard error over 10 iterations of the deploying error.

### 5.2.2 Optimal training horizon

Figure 12 depicts the deploying errors $\mathrm{e}_{\mathrm{deploy}}^k$ over increasing training horizons $h_k \in H$ for the LSTM and FullyConv.

For GC4, Figure 12(a) shows significant error drops for $h_{\mathrm{train}} = 3$ which indeed corresponds to the mixing time of the system $t_{\mathrm{mix}} = 3$. This means that both learning models gain accuracy as soon as some experience of the steady state is recorded in the training trajectories. However, few more training time steps are needed to reach very low loss when $h_{\mathrm{train}} \geq 6$. Remarkably, there are not evident differences between the two learning models used. To better understand the mixing impact on the model learning, we compare the deploying and test errors of the LSTM for different training horizons in Figure 13. Before the mixing time for $h_{\mathrm{train}} = 2$, no experience of the steady state is stored in the training set. Thus, all the predictions are solely based on the influence experienced in the first 2 time steps. As a consequence, Figure 13(a) shows the overfitting effect: the test error is improving over increasing training epochs and tending quickly to zero as the approximations are very accurate for early time steps; on the other hand, the deploying error becomes larger for increasing training epochs. After mixing, for $h_{\mathrm{train}} = 4$ the deploying error has a significant decrease.

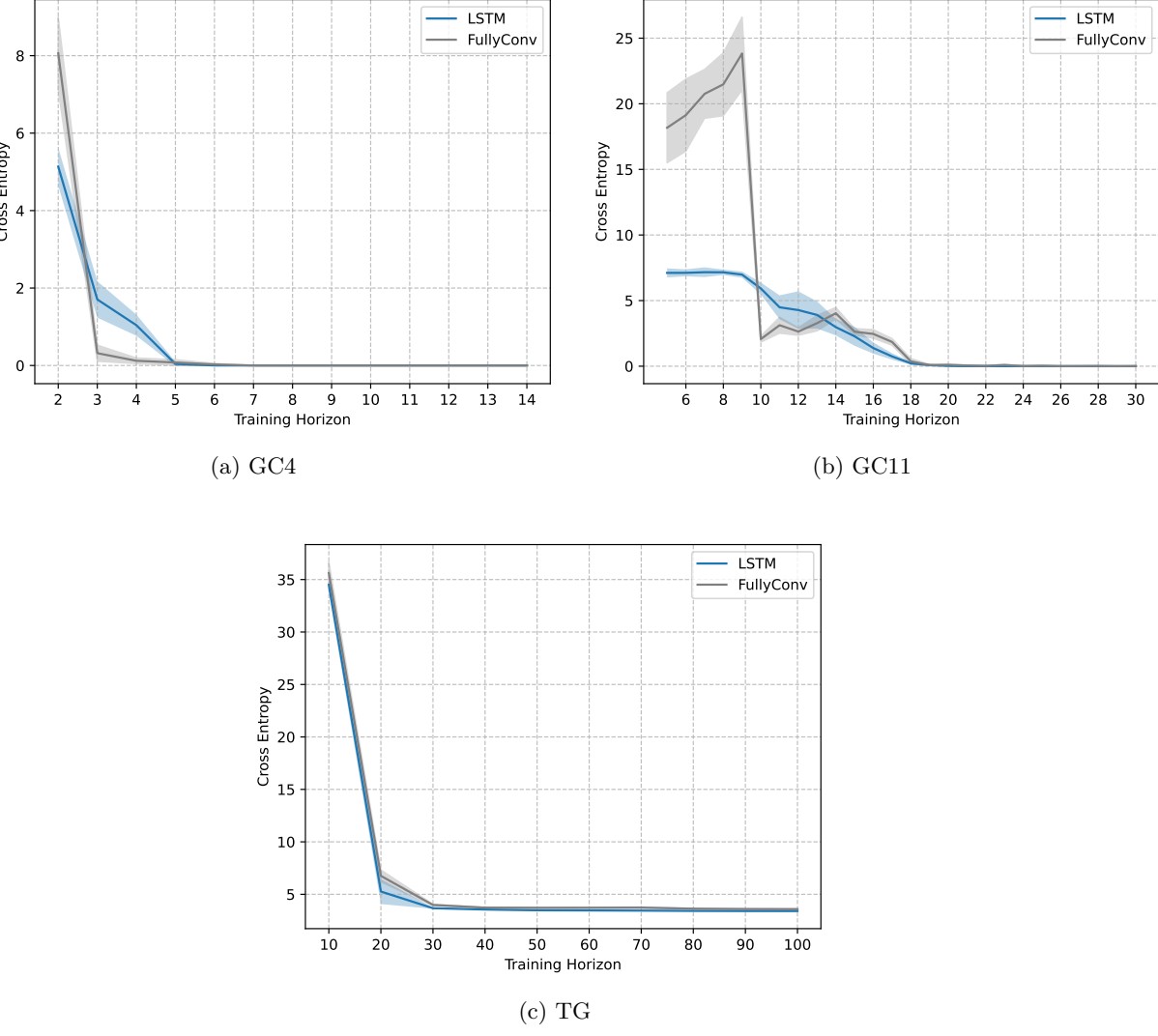

(a) GC4

(b) GC11

(c) TG

Figure 12: Deploying error over training horizons.

Yet Figure 13(b) shows how more training samples worsen the long term predictions. The reason is that the steady-state experience in the training sequences is still not sufficient to be generalized by the learning model over long time steps. Instead, after 6 time steps, the generalization ability of the learning model over the entire deploying horizon improves along with training epochs, as shown in Figure 13(c). Moreover, no significant differences are detected for additional training time steps (see Figure 13(d) for $h_{\text{train}} = 8$). This motivates the choice of training horizon as $h_{\text{train}}^* = 6$.

Similarly for GC11, Figure 12(b) shows that the deploying errors of both learning models decrease consistently when the length of the training sequences exceeds the mixing time $t_{\text{mix}} = 10$. Then, the models need approximately additional 10 time steps more to have a sufficient experience of the stationary influence and converge to a deploying error close to zero when $h_{\text{train}} \geq 20$. More training time steps do not improve significantly the predictions. We select as training horizon $h_{\text{train}}^* = 22$. See also the curves in Figure 16 in Appendix A.

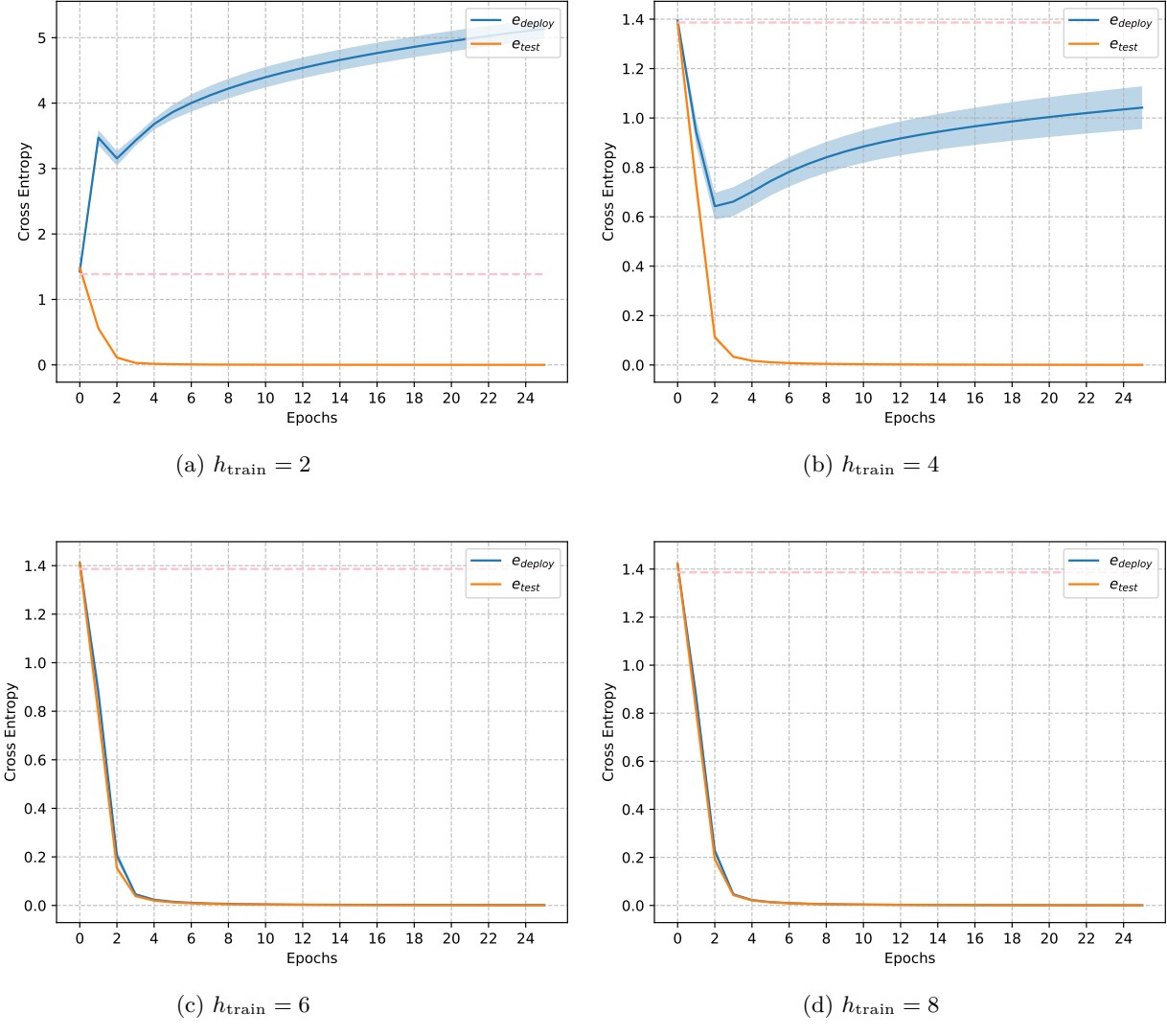

(a) $h_{\text{train}} = 2$        (b) $h_{\text{train}} = 4$

(c) $h_{\text{train}} = 6$        (d) $h_{\text{train}} = 8$

Figure 13: Deploying and test errors for LSTM in GC4 scenario. The dashed line represent the baseline accuracy of the random classifier.

In essence, the experiments in both Grab a chair scenarios demonstrate that regardless of the learning model, the predictions over the entire deploying horizon improve substantially when the training horizon is longer

than the mixing time. Then, learning models require few additional training time steps to acquire the sufficient experience of the steady state and generalize it over the deploying horizon. Much longer training horizons do not bring significant improvements. Conversely, running unnecessarily long global simulations may undermine the advantages of the entire abstraction method. It is therefore crucial to find a good candidate training horizon $h^*_{\text{train}}$.

In Figure 12(c) for the traffic domain, we see the deploying error of both learning models decreasing substantially between 10 and 20 time steps. The performance keeps improving for $h_{\text{train}}$ between 20 and 30 time steps. Increasing further the training horizon does not seem to be beneficial, as also shown by the test and deploying errors collected in Figure 17 in Appendix A. This might also intuitively suggest that the system reaches stationarity between 10 and 30 time steps. Even in this case, the LSTM and FullyConv perform essentially similarly: high errors before mixing time, a decreasing trend after mixing to stabilize around the same training horizon. That leads us to conclude that the best suitable training horizon $h^*_{\text{train}}$ is approximately independent on the learning model but strongly depends on the mixing time of the system.

### 5.2.3 Estimating the deploying error

The results in the last section show that we can find an optimal training horizon $h^*_{\text{train}}$ as the minimum horizon in the set $H$ which guarantees a small deploying error. Nonetheless, how to find such horizon by using only relatively short training sequences still remains an open question. In fact, we based our choice of $h^*_{\text{train}}$ on the deploying error computed over much longer trajectories. However, in practical situation, we only dispose of the short global trajectories to learn the influence. Yet good estimators of the deploying error can still lead the optimal training horizon search. Thus, we want to investigate error measures suitable to estimate the deploying error using test trajectories with length at most as the training horizon.

Supported by the empirical evidence presented in Section 5.2.2, we assume that the system mixes fast in the deploying horizon scale and the learning models require few additional training time steps after mixing to get a quasi-steady representation of the stationary influence, i.e. $t_{\text{mix}} \leq h^*_{\text{train}} \ll h_{\text{deploy}}$. For any $h \geq h^*_{\text{train}}$ the error function can be decomposed as

$$e^k(h) = \frac{1}{h} \left[ \sum_{t=1}^{h^*_{\text{train}}-1} \hat{\text{CE}}(I^t, \hat{I}^t_k) + \sum_{t=h^*_{\text{train}}}^{h} \hat{\text{CE}}(I^t, \hat{I}^t_k) \right]. \tag{10}$$

In the formula, we distinguish between short and long time scales: when $t < h^*_{\text{train}}$, before the system has mixed and the model has learned a stable representation of the steady-state influence, and for $t \geq h^*_{\text{train}}$ when $\hat{\text{CE}}(I^t, \hat{I}^t_k)$ are approximately constant terms representing the model's error on the stationary influence. Given that $h_{\text{deploy}} \gg h^*_{\text{train}}$, the deploying error is mainly affected by the errors in the long time scale, while the contribution of the first term of equation 10 is negligible. On the other hand, the errors on the short time scale have a relevant impact on the test error for a training horizon $h_{\text{train}} \ll h_{\text{deploy}}$. Such terms characterizing the test error give no insight on the model errors after an approximately steady representation of the stationary influence is reached. For these reasons, we do not expect that the test error provides a good estimate of the deploying error. Instead we introduce the test-tail error as defined in equation 8 which we show to approximate better the deploying error when the test set includes some experience of the stationary influence, i.e. for $h_{\text{train}} \geq t_{\text{mix}}$.

The results presented in Figure 14 confirm the theoretical intuition. In the Grab a chair scenarios, all agents start by deterministically choosing the chair on their right side, to then copy the action of their neighbor agent. Thus, agent $N$ keeps picking that chair for the first $t_{\text{mix}} = N - 1$ time steps, to then copy the action of the local agent according to equation 9. Figure 14(a) and 14(b) show that the learning model succeeds well in capturing the two deterministic distributions of the influence sources $a^t_N$ before and after mixing, i.e. test errors are close to zero. However, when $h_{\text{train}} < t_{\text{mix}}$ the training set only includes experience of agent $N$ picking the right side chair. Therefore, the influence model forecasts this behavior indefinitely, thus incurring in a consistently high deploying error. After mixing, the deploying error rapidly tends to zero according to the growing experience of the $N$ agent copying the local actions in the training set. Unlike the test error, for $h_{\text{train}} \geq t_{\text{mix}} - 1$, the test-tail error reflects the deploying error trend since it corresponds exactly to the model error over the stationary influence.

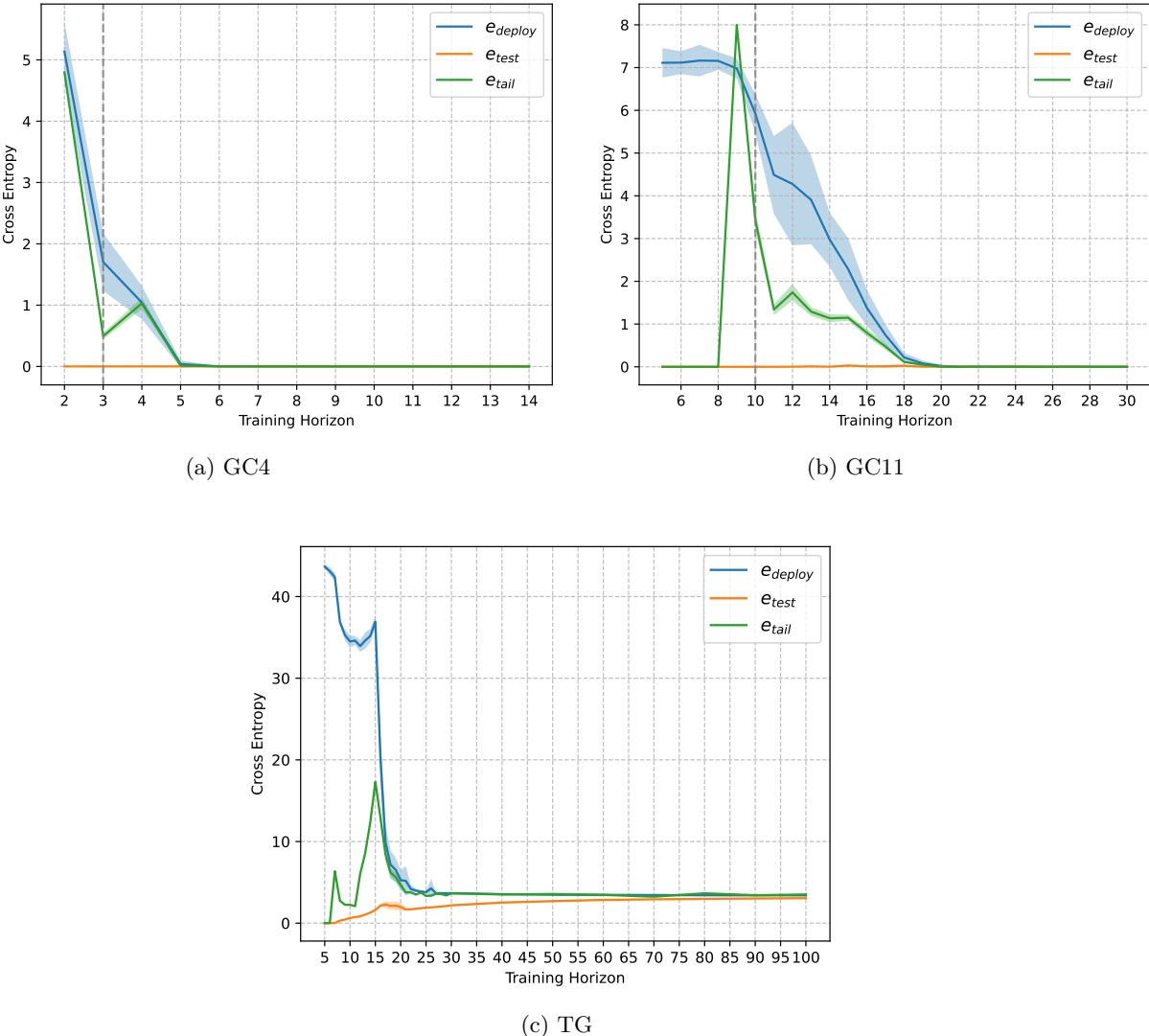

(a) GC4

(b) GC11

(c) TG

Figure 14: Deploying, test and tail-test error for increasing training horizon. The vertical dashed line marks the mixing time of the system, if known.

Figure 14(c) depicts a similar situation in the traffic domain. For short training horizons, the deploying error is significantly high compared to the low values of the test error. Such large difference is explained by the fact that the initial distribution of the system is far from representing the equilibrium: the vehicles enter stochastically in an initially empty road network. Therefore, the model is learning well the distribution of sparse vehicles but it has no experience, for instance, of car jams occurring at later busier time steps. Instead, after approximately 15 initial time steps, the test-tail error closely follows the deploying error behavior. One convincing explanation is that the system mixes before 15 time steps. After that, the short test trajectories are already representative of the steady-state behavior of the influence sources, so is the test-tail error. Moreover, observing the test error curve we notice that very low values for short training horizons, meaning that the model effectively predicts the initial deterministic distributions of the influence sources. The test error progressively increases together with the rise in the system's stochasticity.

Supported by the theoretical intuition and the experimental results, we can derive the following conclusions. In general, the model's test error does not provide insight on the deploying error. We have distinguished

between different ranges of the training horizon. For $h_{\text{train}} < t_{\text{mix}}$, the influence sources have not reached a stationary distribution, the test-tail error is not meaningful and the learning models have typically high deploying error. When $t_{\text{mix}} \leq h_{\text{train}} \leq h_{\text{train}}^*$, the model predictions of the steady behavior of the influence are not accurate and stable, the deploying error starts decreasing along with the experience of stationarity collected in the test sample. In this range, the test-tail error is particularly useful to estimate the deploying error since gives an approximation of the model error over the stationary influence point. For $h_{\text{train}} \geq h_{\text{train}}^*$, the model provides good and steady representations of the stationary influence. The test, test-tail and deploying errors asymptotically converge to a limit cross entropy value which is lower bounded by the entropy of the stationary influence point. Therefore, the test-tail error can be employed to determine the training horizon $h_{\text{train}}^*$, as the lowest horizon after mixing, which minimizes the test-tail error. This argument together with the results presented in Figure 14, fully explain our choices of $h_{\text{train}}^* = 6$ for GC4, $h_{\text{train}}^* = 22$ for GC11 and $h_{\text{train}}^* = 30$ for TG.

### 5.2.4 Observations

In summary, our experiments show that

- Learning models can generalize influence points beyond the training horizon.

- For each scenario, a minimal training horizon $h_{\text{train}}^*$ suitable to learn effectively the influence for much longer horizons $h_{\text{deploy}}$ can be found. This number depends on the dynamics of the system but it is approximately independent of the learning model.

- Neglecting the contributions of early time steps cross entropy errors in the computation of the test error allows to obtain a good estimate of the error over horizon $h_{\text{deploy}}$ that can be used to determine the optimal training horizon $h_{\text{train}}^*$.

## 6 Conclusions

In this paper we investigate learning models and techniques for the influence learning task in realistic scenarios. We run an extensive empirical investigation of the performance of different learning models in a variety of domains. We conclude that complex scenarios may still induce manageable influence learning task. Relatively small recurrent models can achieve the same performance levels as state-of-the art fully convolutional neural networks. Moreover, we explore how to leverage learning models to build local simulators for long horizons using short training trajectories. In particular, we show that there exists a training horizon which is sufficient to learn good influence approximations for long horizon problems. This horizon strongly depends on the mixing properties of the system. On the other hand, it is essentially independent of the learning model. Finally we show how to leverage a suitable test error to determine this optimal training horizon.

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

# A    Additional results

In Figure 15 we provide the learning curves for the experiments presented in Section 4 for the System admin domain.

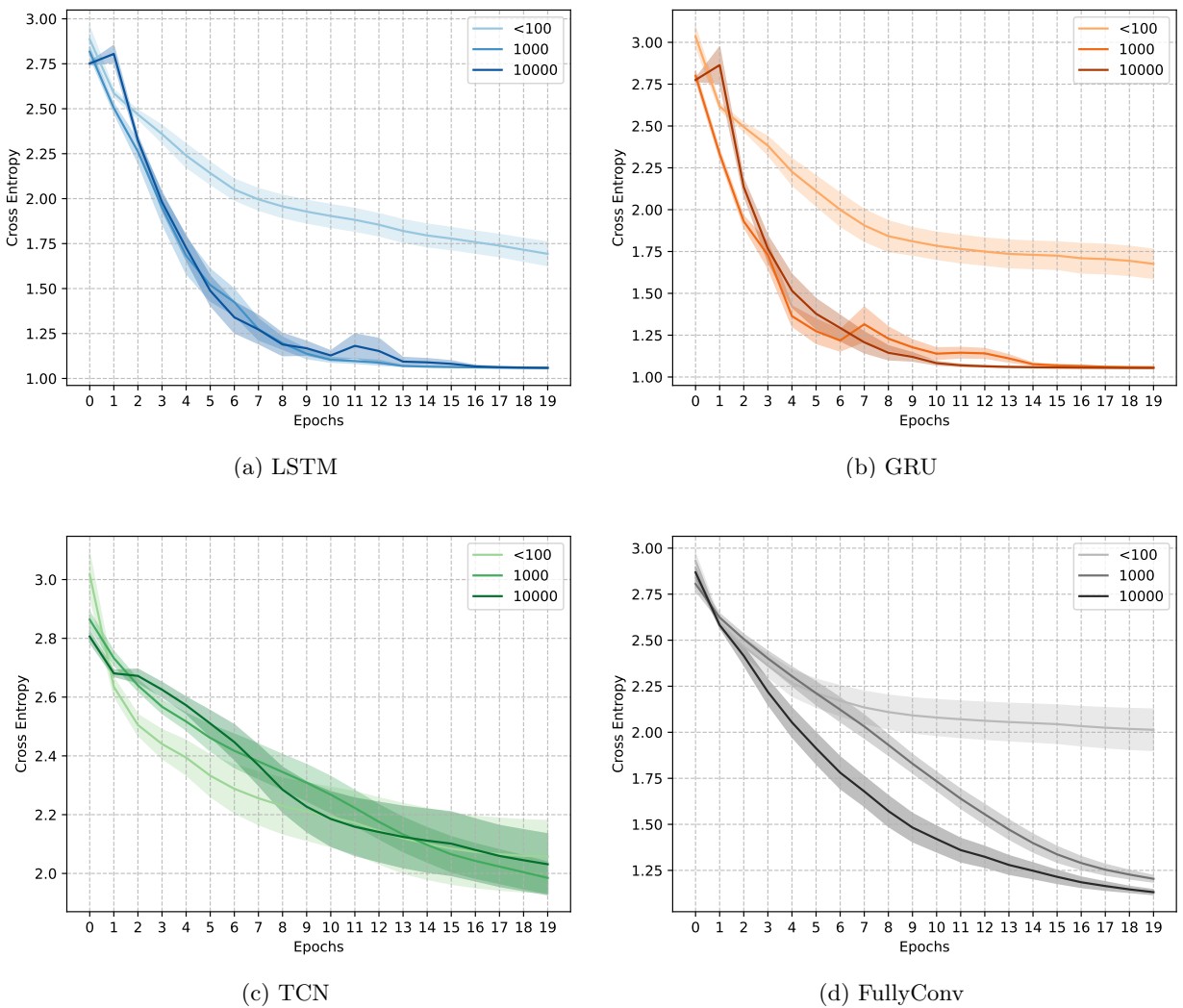

(a) LSTM

(b) GRU

(c) TCN

(d) FullyConv

Figure 15: System admin. Cross entropy test errors over epochs.

Figure 16 shows the test and deploying errors of an LSTM learning model for the influence over the different training horizons in GC11 domain for the experiments in Section 5.

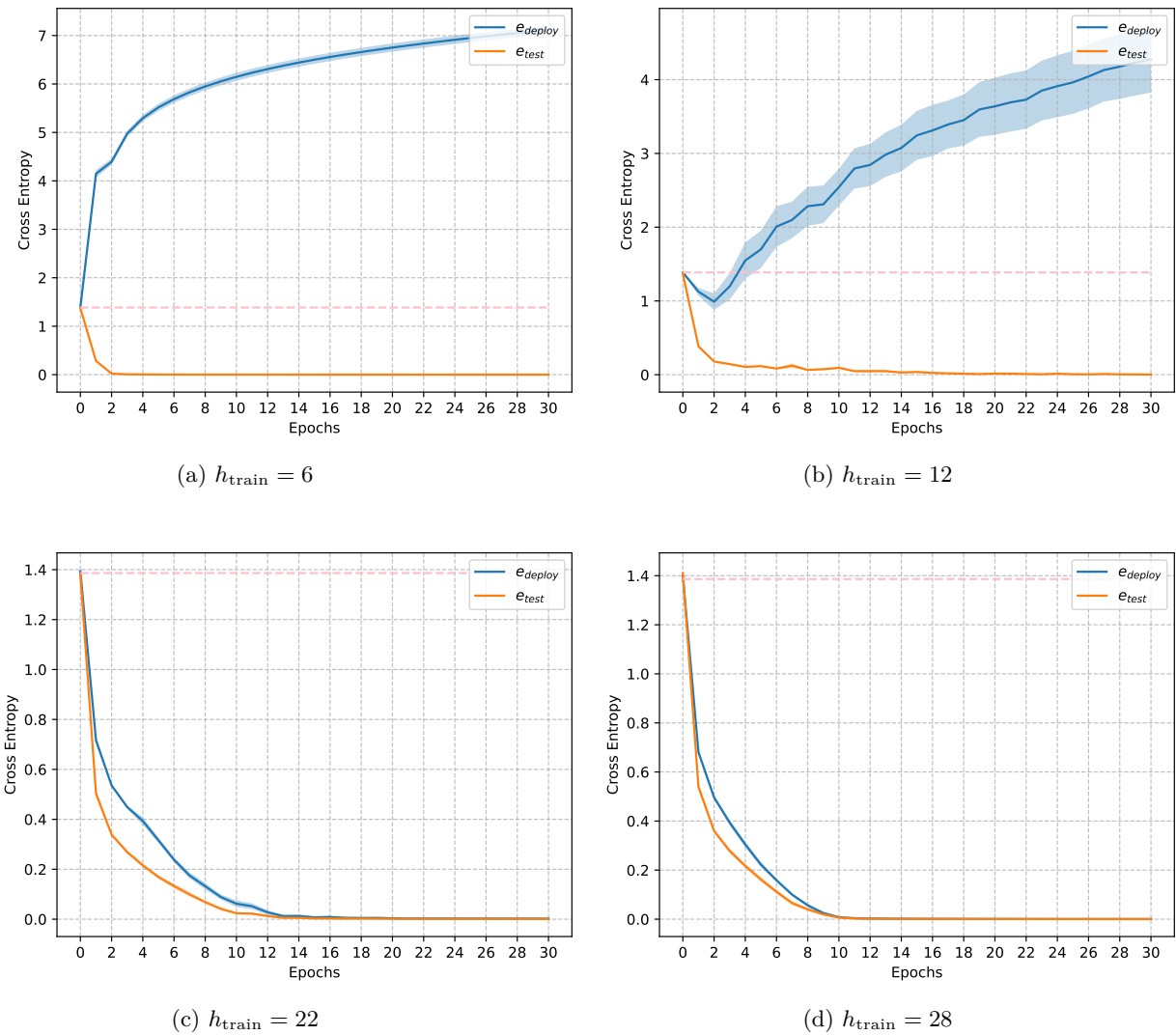

(a) $h_{\text{train}} = 6$

(b) $h_{\text{train}} = 12$

(c) $h_{\text{train}} = 22$

(d) $h_{\text{train}} = 28$

Figure 16: Deploying and test errors for LSTM in GC11 scenario. The dashed line represent the baseline accuracy of the random classifier.

Figure 17 shows the test and deploying errors of an LSTM learning model for the influence over the different training horizons in TG domain for the experiments in Section 5

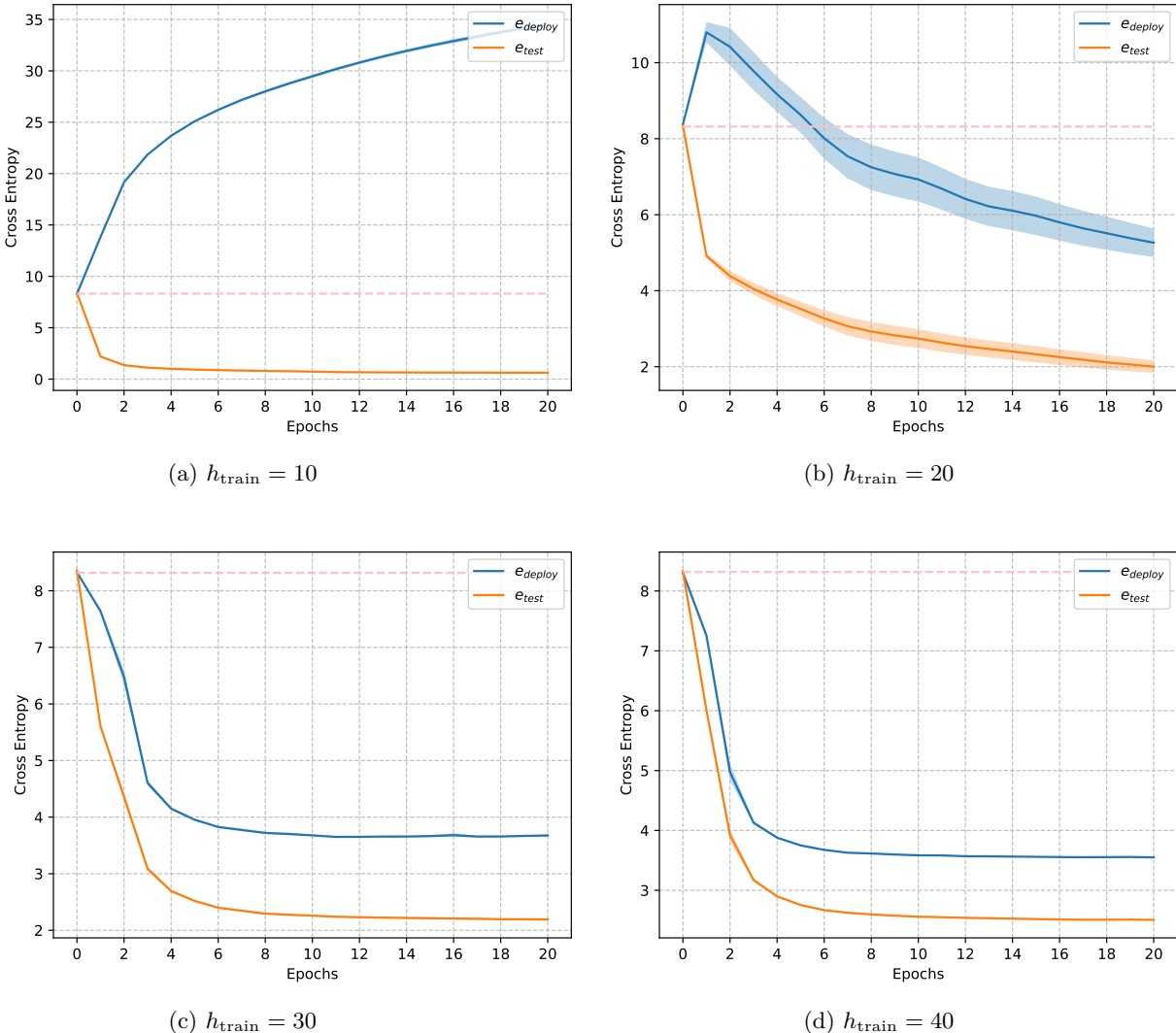

(a) $h_{\text{train}} = 10$

(b) $h_{\text{train}} = 20$

(c) $h_{\text{train}} = 30$

(d) $h_{\text{train}} = 40$

Figure 17: Deploying and test errors for LSTM in TG scenario. The dashed line represent the baseline accuracy of the random classifier.

## B  Experimental setup

| Domain | $h$ | #Agents | Policies | $b_0$ | #Influence Sources | D-set dimension |
|--------|-----|---------|----------|-------|---------------------|------------------|
| Microgrid | 40 | 100 | ranges | uniform | 4 | 1 |
| Traffic grid | 100 | 9 | priority | zeros | 12 | 9 |
| System admin | 500 | 20 | mixed | uniform | 4 | 3 |

Table 5: Settings of the scenarios and features of the influence learning problem.

| Domain | Optimization | | | | | | |
|--------|-------------|-----------|---------|-----|------|----------|-------|
| | Sample size | Batch size | #Epochs | Alg | Loss | LR Decay | Valid |
| Microgrid | 500 | 100 | 15 | Adam | Entropy | Linear | Split90% |
| Traffic grid | 500 | 100 | 20 | Adam | Entropy | Linear | Split90% |
| System admin | 500 | 100 | 20 | Adam | Entropy | Linear | Split90% |

Table 6: Optimization hyperparameters for the deep learning models.

| Models | Architecture | | | | | |
|--------|---------|--------|--------|---------|----------|------------|
| | #Layers | #Units | Kernel | #Params | Activate | Regularize |
| *(size ≤100)* | | | | | | |
| LSTM | 1 | 2 | - | 88 | Tanh | None |
| GRU | 1 | 2 | - | 78 | Tanh | None |
| TCN | 2 | 2,2 | 8 | 100 | ReLU | None |
| FullyConv | 2 | 1 | 8 | 52 | ReLU | Dropout |
| *(size 1000)* | | | | | | |
| LSTM | 1 | 13 | - | 1056 | Tanh | None |
| GRU | 1 | 14 | - | 954 | Tanh | None |
| TCN | 4 | 6,6,6,6 | 8 | 1048 | ReLU | None |
| FullyConv | 4 | 6,6 | 8 | 1084 | ReLU | Dropout |
| *(size 15000)* | | | | | | |
| LSTM | 1 | 56 | - | 14128 | Tanh | None |
| GRU | 1 | 64 | - | 13904 | Tanh | None |
| TCN | 5 | 20,20,20,20,20 | 8 | 13396 | ReLU | None |
| FullyConv | 8 | 15,15,15,15 | 8 | 13246 | ReLU | Dropout |
| LogReg | - | - | - | 13104 | None | None |

Table 7: MicroGrid. Architectures of the deep learning approaches for the comparison of learning models.

| Models | Architecture | | | | | |
|---|---|---|---|---|---|---|
| | #Layers | #Units | Kernel | #Params | Activate | Regularize |
| *(size ≤200)* | | | | | | |
| LSTM | 1 | 2 | - | 176 | Tanh | None |
| GRU | 1 | 2 | - | 150 | Tanh | None |
| TCN | 2 | 2,2 | 4 | 164 | ReLU | None |
| FullyConv | 2 | 2 | 4 | 188 | ReLU | Dropout |
| *(size 1000)* | | | | | | |
| LSTM | 1 | 9 | - | 960 | Tanh | None |
| GRU | 1 | 11 | - | 1014 | Tanh | None |
| TCN | 4 | 4,4,4,4 | 10 | 976 | ReLU | None |
| FullyConv | 4 | 4,4 | 10 | 1032 | ReLU | Dropout |
| *(size 10000)* | | | | | | |
| LSTM | 1 | 42 | - | 9936 | Tanh | None |
| GRU | 1 | 49 | - | 10020 | Tanh | None - |
| TCN | 4 | 16,16,16,16 | 10 | 9592 | ReLU | None |
| FullyConv | 4 | 16,16 | 10 | 9816 | ReLU | Dropout |
| *(size 50000)* | | | | | | |
| LSTM | 1 | 104 | - | 50360 | Tanh | None |
| GRU | 1 | 120 | - | 50064 | Tanh | None - |
| TCN | 6 | [30]x6 | 10 | 48624 | ReLU | None |
| FullyConv | 6 | [30,30,30] | 10 | 49104 | ReLU | Dropout |
| *(size 1M)* | | | | | | |
| LogReg | - | - | - | 1093200 | None | None |

Table 8: TrafficGrid. Architectures of the deep learning approaches for the comparison of learning models.

| Models | Architecture | | | | | |
|---|---|---|---|---|---|---|
| | #Layers | #Units | Kernel | #Params | Activate | Regularize |
| *(size ≤100)* | | | | | | |
| LSTM | 1 | 2 | - | 80 | Tanh | None |
| GRU | 1 | 2 | - | 66 | Tanh | None |
| TCN | 2 | 2,2 | 4 | 68 | ReLU | None |
| FullyConv | 2 | 2 | 4 | 80 | ReLU | Dropout |
| *(size 1000)* | | | | | | |
| LSTM | 1 | 12 | - | 920 | Tanh | None |
| GRU | 1 | 14 | - | 918 | Tanh | None |
| TCN | 4 | 6,6,6,6 | 8 | 1088 | ReLU | None |
| FullyConv | 4 | 6,6 | 8 | 1136 | ReLU | Dropout |
| *(size 10000)* | | | | | | |
| LSTM | 1 | 48 | - | 10568 | Tanh | None |
| GRU | 1 | 54 | - | 9998 | Tanh | None |
| TCN | 6 | [16] x6 | 8 | 10856 | ReLU | None |
| FullyConv | 6 | 16,16,16 | 8 | 11016 | ReLU | Dropout |
| *(size 3M)* | | | | | | |
| LogReg | - | - | - | 2.9M | None | None |

Table 9: System admin. Architectures of the deep learning approaches for the comparison of learning models.

| Domain | $H$ | $h_{\text{deploy}}$ | #Agents | Policies | $b_0$ |
|--------|-----|------|---------|----------|-------|
| GC4 | $\{2, \ldots, 14\}$ | 200 | 4 | copy | Deterministic - right chair |
| GC11 | $\{5, \ldots, 30\}$ | 200 | 11 | copy | Deterministic - right chair |
| TG | $\{10, \ldots, 100\}$ | 500 | 9 | priority | Deterministic - zero vehicles |

Table 10: Scenario settings for the influence in long horizon tasks.

| Domain | Batch size | #Epochs | LR Init | LR Final | LR Decay | Train size $n$ | Test size $m$ |
|--------|-----------|---------|---------|----------|----------|----------------|---------------|
| GC4 | 10 | 25 | $10^{-2}$ | $10^{-5}$ | linear | 500 | 100 |
| GC11 | 10 | 30 | $10^{-2}$ | $10^{-5}$ | linear | 500 | 100 |
| TG | 10 | 20 | $10^{-2}$ | $10^{-5}$ | linear | 500 | 100 |

Table 11: Optimization choices for the influence in long horizon tasks.

| Domain | Model | Architecture | | | | | |
|--------|-------|---------|--------|--------|---------|----------|------------|
| | | #Layers | #Units | Kernel | #Params | Activate | Regularize |
| GC4 | LSTM | 1 | 10 | - | 564 | Tanh | None |
| | FullyConv | 4 | [6,6] | 4 | 544 | ReLU | Dropout |
| GC11 | LSTM | 1 | 32 | - | 4612 | Tanh | None |
| | FullyConv | 8 | [10,10,10,10] | 6 | 4484 | ReLU | Dropout |
| TG | LSTM | 1 | 32 | - | 2136 | Tanh | None |
| | FullyConv | 4 | [8,8] | 6 | 1944 | ReLU | Dropout |

Table 12: Architectures of the deep learning approaches for the influence in long horizon tasks.

