# OpenReview forum: "Influence Learning in Complex Systems"
_TMLR — Rejected by TMLR_

### Review · Reviewer_Rj5H · 2023-06-08

**Summary Of Contributions:**

The authors are interested in complex environments where the global dynamics, which verifies Markov property, can be represented as a set of interactions between subparts of the environment. From the standpoint of a subpart, the dependencies to the other subparts do not verify Markov property and the control policy over each subpart must leverage an approximate representation of these dependencies (the influence learning problem). In a first experimental study, using 3 environments, the authors investigate the impact of various architectures used to model this influence and of their size. They conclude that the influence learning problem is relatively easy and that small enough recurrent models provide accurate influence learning. In a second experimental study, they focus on the case where dependencies are exerted over a long horizon, and they investigate whether architectures capturing dependencies at a shorter term can be good enough at modelling these cases. They conclude that an optimal short term can easily be found and that indeed this can be enough even for capturing long term dependencies.

**Audience:**

Yes

**Broader Impact Concerns:**

I don't see any direct broader impact concern beyond the ones usually associated with RL research on the control of complex systems.

**Claims And Evidence:**

Yes

**Requested Changes:**

The paper is too long. I see two ways to mitigate this issue:

- consider that there should be two papers, corresponding to the two studies that I outlined in my summary. It seems to me that these two studies are close to independent (the second study does not build much on the conclusions of the first study)

- try to write a much shorter, message-oriented and impactful version of the same work. For instance, one could describe the evaluation environments in a few lines and put the long description in appendices. Also, in the first empirical study, one could only put forward in the main part the images that deliver the insights corresponding to the conclusion (mostly, the Pareto fronts and error profiles) and also put all the intermediate results in appendices. Most images are too large, etc. Try to write the same paper in 12 pages and it will get much more interesting.

It would be nice to measure how much suboptimal one gets using the influence learning approach on the studied benchmarks with respect to brute force solution of the global MDP. If this study has been performed before in other papers (the team conducting these works is easy to identify given the 9 self-references), the authors could still remind this performance as an upper-bound of what they can get and maybe also use a random-like, dumb solution to lower-bound the performance.

Open question: what if the variables are not binary? Can you extend the 2TBN formalism to deal with environments shoing scalar or continuous variables?

Small issues:

- the environments and the rationale for their choice are described several times, any repetition should be avoided

- the choice of highlighted results in Tables 1, 2 and 3 is only explained late, it should be explained in the caption of these tables, as it does not correspond to the most standard choice (highlighting the best performances).

Language issues:

- Markovianity is probably a neologism

- in english, in general, put the adverb before the verb: represent accurately, leverage effectively, affect only, ...

- consist of replacing -> in

- what extend -> extent (?)

- p4. concide Oliehoek et al. (2021) -> use \citep


**Strengths And Weaknesses:**

Weaknesses:

- the main part of the paper is 23 pages long (references excluded) and the same messages could be delivered in half this size (see "requested changes" for advices).

- the english can be improved

Strengths:

- the research methodology looks sound, although to be honest I have to admit that I did not pay much attention to all the details given the unusual length of the paper

---

> ### Author Response · Authors · 2023-07-12
>
> We appreciate the reviewer's comments on trying to make a paper focus, accessible and concise. To this end, we will follow the suggestion to move the domain descriptions to the appendix, resize the images and check for repetitions. However, since no specific instructions with respect to the length of the submissions for TMLR were given and typical length of other journals in the field as JAIR, JMLR is also between 20 - 35 pages, we don’t intend to rewrite a new shorter version of the paper. Also, we believe that splitting our paper in two, would lead to a fractured perspective. Specifically, the current paper takes a comprehensive approach towards validating the potential of real-world impact and feasibility of learning approximate influence representations. We feel that both the investigation of different architectures as well as the issue of dealing with long horizons would need to be an integral part of that.
>
> This bound shows that the loss in value is bounded by the maximum KL-divergence error of the influence predictions. This observation motivates us to train learning models that minimize cross-entropy loss as it is well-aligned with the objective of minimizing the maximum KL-divergence error. In the final version, we can provide further clarifications on these aspects.
> Concerning suboptimality of solutions, influence-based abstraction is a lossless abstraction approach. That means that both the brute-force solution of the global problem and the solution obtained by considering local problems with influence yield the same value. However, we use approximate models for the influence and this introduces loss in value. Such loss is upper bounded as described in equation 1 Section 3.1 (see “Loss bounds for approximate influence-based abstraction. (AAMAS), 2021, Congeduti et al”). This shows that the loss in value is bounded by the maximum KL-divergence error of the influence predictions. This observation motivates us to train learning models that minimize cross-entropy loss as it is well-aligned with the objective of minimizing the maximum KL-divergence error. In the final version, we can provide further clarifications on these aspects.
>
> “what if the variables are not binary? Can you extend the 2TBN formalism to deal with environments shoing scalar or continuous variables?”
> The 2DBN formalism ​​includes continuous variables and the definition of influence can be easily extended for continuous probability distributions. However, continuous distributions induce a different learning task which is typically approached by assuming a distribution function class and learning parameters for such distribution (e.g. mean and standard deviation). In our empirical study, we consider only discrete influence sources (MicroGrid influence sources are not binary but discrete) and do not include the continuous case. We acknowledge that this is a limitation of our work particularly as we aim to encompass realistic scenarios. For this reason, we intend to explicitly mention such limitations.
>
> We will make sure to proofread the paper, check for misspellings and improve language.

---

### Review · Reviewer_6pfe · 2023-07-02

**Summary Of Contributions:**

The paper presents a novel approach to reinforcement learning using influence-based abstraction (IBA) to break down complex problems into manageable subproblems. It investigates different learning methods for approximating influence in realistic domains and demonstrates the effectiveness of lightweight local simulators for long horizon tasks. The authors provide empirical evidence and theoretical insights to support their approach, showing that learning influence approximations is generally easier and requires fewer samples than solving the reinforcement learning or planning problem.

**Audience:**

Yes

**Claims And Evidence:**

Yes

**Requested Changes:**

**Questions**

1. Could the authors provide a more precise definition of influence learning, preferably in a mathematical form? My understanding is that influence learning aims to identify and leverage the correlations between agents, thereby allowing for the disregard of agents with low influence strength on a given agent to enhance learning efficiency. However, a more formal definition would be beneficial for clarity.

2. Could the authors elaborate on the relationship between influence learning and other methods such as attention mechanisms or mean-field methods? The paper does not explicitly discuss these relationships, and further clarification would be valuable for understanding the context and potential advantages of the proposed approach.

3. Could the authors elaborate on the challenges associated with influence learning for long-horizon tasks? The paper mentions the application of influence learning in this context, but a more detailed discussion on the specific difficulties encountered would be beneficial for a comprehensive understanding.

**Other Suggestions**
1. Discussion on Limitations: Provide a more detailed discussion on the limitations of the proposed method. Every method has its limitations, and discussing them helps to provide a balanced view and directions for future work.
2. Implementation Details: Include more details about the practical aspects of implementing the proposed method. This would be particularly helpful for readers who wish to replicate your work or build upon it.
3. Scalability Analysis:  Provide an analysis of how well the proposed method scales to larger, more complex problems. This would help to demonstrate the applicability of your method in a wider range of scenarios
4. Code Availability: If possible, consider providing the code used for your experiments. This would greatly aid reproducibility and allow others in the community to build upon your work.

**Strengths And Weaknesses:**

**Strengths**

1. The paper presents a novel approach to reinforcement learning using influence-based abstraction (IBA) to break down complex problems into manageable subproblems. This is a significant contribution to the field.

2. Empirical Investigation: The authors conduct an empirical investigation of different learning methods for approximating influence in realistic domains. This provides practical insights into the effectiveness of their proposed method.


**Potential Weaknesses**

See 'Requested Changes' Section.

---

> ### Author Response · Authors · 2023-07-12
>
> We thank the reviewer for the suggestions and questions raised.
>
> Question 1 - For a fixed set of policies of external agents \pi_{-i}, the influence that those agents exert on the local model of a protagonist agent i, is the conditional probability distribution of the influence sources s_{src} (external factors affecting directly the local model) given the local factors history (x^0,...,x^t) which separates the local variables from the external ones, i.e. P(s^t_{src} | x^0,...,x^t ,\pi_{i}) =: I (s_{src} | x^0,...,x^t). The objective of influence learning is the probability distribution I. Namely, we want to learn an approximate \hat I which given the local factors history (x^0,...,x^t) as input, outputs a probability distribution over the influence sources s_{src}, i.e. \hat I (x^0,...,x^t) \in \Delta (s^t_{src}). We will make sure to add rigorous definitions for influence and influence learning problem in Section 2 and 3 to clarify the formal setting.
>
> Question 2 - The work “Suau, Miguel, et al. "Influence-aware memory architectures for deep reinforcement learning in POMDPs." Neural Computing and Applications (2022)”  relates influence based abstraction to attention. In particular, a spatial attention mechanism is used to select the relevant information in the local observation for the agent’s problem. Here, attention serves to filter out from the d-set the local variables which do not add necessary information.
> Under the assumption that a high number of anonymous agents influences one influence source that is not affected by the protagonist agent, the influence can be interpreted as the mean-field occupancy distribution that captures the effect of the external agents on the protagonist's local problem. We can make sure to explicitly refer to these in the paper.
>
> Question 3 - In tasks which last for many time steps, collecting trajectories using the large simulators for the entire horizon might be infeasible in practice. Instead, it would be required to learn models of influence from short trajectories that generalize well to the entire deployment horizon. Whether and in which cases this is possible and effective, is not completely apparent. We investigate this by training influence models over short trajectories and assessing the approximations when deployed for long horizons. We can elaborate slightly more on this.
>
> Other suggestions -
> We will explicitly discuss limitations and assumptions of our work. For instance, related to the choices of the domains which are inspired by realistic situations but still represent simplifications of actual real-world scenarios. Additionally, our investigation focuses exclusively on discrete state variables and does not encompass the more realistic problem of learning continuous influence sources. We intend to make the code publicly available upon publication.

---

### Review · Reviewer_MCnv · 2023-07-07

**Summary Of Contributions:**

This paper assumes a large factored problem can be decomposed into several sub-problems and use multiple local lightweight simulators to model the entire environment to reduce the sample complexity. When doing this, the influence approximation among those local simulators is crucial. The focus of this paper is to investigate the influence of different aspects of different network structures on learning the influence approximation.

**Audience:**

Yes

**Broader Impact Concerns:**

Not applicable.

**Claims And Evidence:**

Yes

**Requested Changes:**

The following problems need to be clarified in the revised version:

There are some technical details confusing:

1) In the Background section, the state is decomposed into three components, while $s_{ext}$ is not explained. And seems like the symbol $s_{src}$ is defined as the external state variables, which is confusing too.

2) The reviewer is not convinced that modeling an environment with multiple interacting agents as a POMDP is appropriate when other agents' policies are fixed.

3) The reviewer is confused about the d-set, how did the authors determine the formula of $d_{set}^{t}$?

4) The details of influence modeling and learning are limited, failing to draw the whole picture.

5) The reviewer is concerned about how the influence is modeled and learned in practice, seems like the decomposition of the state should be made manually, which is non-trivial. The motivation of the paper is to show that a complex problem is manageable if the influence can be approximated accurately. However, based on the fact of human effort decomposing the state (space), this may raise another concern, as different people may have different preferences or biases.

**Strengths And Weaknesses:**

Strengths:

This paper proposes an interesting point of view of how machine learning methods solve a large-scale, complex problem in practice.

This paper conducts extensive experiments on several convincing domains.

Weaknesses:

Some technical details need to be clarified.

The assumptions and limitations should be explicitly described.

---

> ### Author Response · Authors · 2023-07-12
>
> We thank the reviewer for pointing out technical aspects that require better clarification. We will address those points as follows.
>
> 1. and 4. - We will incorporate more rigorous definitions in the Background section. Specifically, we will define the different components of the state (external state s_ext, influence sources s_{src}, local state) and the conditional probability distributions of influence sources given the d-set (the influence) which serves as the target of our learning problem. Additionally, we will include a formal definition of input and output of the influence learning problem in Section 3.
>
> 2. - We think that it is relatively common to use POMDP to compute best responses in multiagent settings when the policies of other agents are fixed. See, for instance:
> Nair, Ranjit, et al. "Taming decentralized POMDPs: Towards efficient policy computation for multiagent settings." IJCAI. Vol. 3. 2003
> Hansen, Eric A., Daniel S. Bernstein, and Shlomo Zilberstein. "Dynamic programming for partially observable stochastic games." AAAI. Vol. 4. 2004
> Oliehoek, Frans A., and Christopher Amato. "Best response Bayesian reinforcement learning for multiagent systems with state uncertainty." Proceedings of the Ninth AAMAS Workshop on Multi-Agent Sequential Decision Making in Uncertain Domains (MSDM). 2014.
>
> 3. -  The d-set is the set of local factors that d-separates the local to the external factors according to the definition of d-separation for causal graph in a 2DBN (see “Christopher M Bishop. 2006. Pattern Recognition and Machine Learning. Springer.”). We will ensure to add this reference to the text. Further details on how this is used in IBA are given by Frans Oliehoek et all. "A sufficient statistic for influence in structured multiagent environments". Journal of Artificial Intelligence Research (JAIR), 2021.
>
> 5. - In our work, we assume that the decomposition into local and external factors is given and we do not aim to discuss efficiency or effects of different possible choices for such decomposition. This might be indeed seen as a potential limitation as it might not be trivial to determine a good trade off between the amount of information to include in the local model and local model size. In practice, however, we have usually found it relatively easy to determine sets of local variables that led to good performance.
>
> We will add other limitations and assumptions of our work. For instance, related to the choices of the domains which are inspired by realistic situations but still represent simplifications of actual real-world cases. Also, we limit our investigation to discrete state variables and do not assess the more realistic problem of learning continuous influence sources.

---

### Review · Reviewer_JruB · 2023-07-10

**Summary Of Contributions:**

This paper studies how large factored problems can be decomposed into smaller subproblems. A global simulator of a system (e.g traffic system or electrical grid) is decomposed into local lightweight simulators with few local state variables. This approach requires modelling the influence of the rest of the whole system on a local part. In this paper the authors investigate a number of ways to learn these influences.

**Audience:**

Yes

**Claims And Evidence:**

Yes

**Requested Changes:**

Questions:
* Is it possible to quantify how suboptimal the found solutions are relative to a brute-force solution of the global problem?

Suggestions:
* Consider shortening the paper. My advice would be to move some of the discussions to the appendix and keep the main paper shorter and more focused. I would already foreshadow main results at the end of the intro.
* Explicitly discuss limitations of the method. It seems that e.g. the state decomposition is done manually. In this context it would also be interesting to see how well the method scales with problem size.
* Figure 9: perhaps the colors in the subfigure could be chose to be slightly more distinguishable.
* deploying error -> deployment error?
* experimental


**Strengths And Weaknesses:**

Strengths:
* The paper nicely motivates an interesting problem setting
* Extensive experimental results on relevant domains.
* as far as I can tell technically sound

Weaknesses:
* I think the paper is too long. Some of the writing could be sharpened and condensed.

---

> ### Author Response · Authors · 2023-07-12
>
> We want to thank the reviewer for the suggestions on improving the focus of the paper.
> In response to the comments received, we plan to move the detailed descriptions of experimental domains to the appendix, resize the figures and review the content to identify and eliminate any instances of repetition. We will also elaborate on the assumptions, such as the local-external decomposition, and discussing the limitations such as transferability of results to actual real-world scenarios.
>
> “Is it possible to quantify how suboptimal the found solutions are relative to a brute-force solution of the global problem?”
>
> Influence-based abstraction is a lossless abstraction approach. Both the brute-force solution of the global problem and the solution obtained by considering local problems with influence yield the same value. However, we use approximate models for the influence and this induces suboptimality. The loss in value from using approximate influence is upper bounded, see equation 1 Section 3.1 as discussed in “Loss bounds for approximate influence-based abstraction. (AAMAS), 2021, Congeduti et al”. This bound shows that the loss in value is bounded by the maximum KL-divergence error of the influence predictions. This observation motivates us to train learning models that minimize cross-entropy loss as it is well-aligned with the objective of minimizing the maximum KL-divergence error. In the final version, we can provide further clarifications on these aspects.

---

### Decision · Action_Editors · 2023-08-29

**Recommendation:** Reject

**Comment:**

As explained above, the main problem with the paper is a lack of clarity regarding the notion of local models and influence learning.  The authors promised to address this in the final version of the paper instead of submiting a revised version during the rebuttal period.  This is unfortunate, since the authors should have revised the paper during the rebuttal.

The paper discusses formalisms for POMDPs and RL, but the goal is not to do control.  The paper should provide the formalism for the decomposition of state variables into local, source and external variables as well as the notion of influence learning.  Since the decomposition is done manually, a recipe for practitioners to follow should be provided, otherwise, one could argue that the experiments are inconclusive since different decompositions may be advantageous/disadvantageous.  Can you specificy formal conditions under which the decomposition into local models will be advantageous and formal conditions under which working with a large global model will be advantageous?

As pointed out by the reviewers, the paper is long and repetitive in some sections.  While there is no page limit, the reader expects a certain amount of insights per page.  At 23 pages, the paper is roughly 3 times the length of an 8-page conference paper, but it does not contain more insights than a conference paper.  Hence, the authors should rework the paper as suggested by the reviewers.

**Audience:**

The work will be of interest to anyone modeling complex temporal processes.

**Claims And Evidence:**

The paper claims that learning complex temporal models by factoring the state space into small local models is advantageous in terms of accuracy and data complexity.  The paper provides an empirical evaluation.  There is no proposed/new algorithm.  Hence, the contribution is the empirical evaluation of the claims.

The main problem with the paper is that it is not clear how small local models are obtained.  As pointed out by the reviewers, there is no formal definition of what constitute a local model nor how the local, source and external variables are obtained/defined.  The notion of influence learning is not made precise either.  The authors acknowledge in the rebuttal that the decomposition is done manually.  Nevertheless, there is a need for a clear and formal definition of those concepts.  The paper claims that factoring the state space into local models is advantageous, but clearly we can imagine some domains where this is the case and other domains where this would not be the case.  Hence, there is a need for clear definitions and criteria that can distinguish when local models will be more accurate and/or can be learnt with less data.

The empirical evaluation is entirely synthetic.  The problems are inspired by real world applications, but since the main contribution is an empirical evaluation, it would be desirable to have at least one real world problem, meaning that the data should be coming from a real application instead of a simulation.  In addition, as pointed out by the reviewers, limitations of the empirical evaluation should be stated.

**Resubmission Of Major Revision:**

The authors may consider submitting a major revision at a later time.